# Learning Compressed Transforms with Low Displacement Rank

**Anna T. Thomas**[†,*] **Albert Gu**[†,*] **Tri Dao**[†], **Atri Rudra**[‡], **Christopher Ré**[†]
[†] Department of Computer Science, Stanford University
[‡] Department of Computer Science and Engineering, University at Buffalo, SUNY
{thomasat,albertgu,trid}@stanford.edu, atri@buffalo.edu, chrismre@cs.stanford.edu

## Abstract

The low displacement rank (LDR) framework for structured matrices represents a matrix through two displacement operators and a low-rank residual. Existing use of LDR matrices in deep learning has applied fixed displacement operators encoding forms of shift invariance akin to convolutions. We introduce a class of LDR matrices with more general displacement operators, and explicitly learn over both the operators and the low-rank component. This class generalizes several previous constructions while preserving compression and efficient computation. We prove bounds on the VC dimension of multi-layer neural networks with structured weight matrices and show empirically that our compact parameterization can reduce the sample complexity of learning. When replacing weight layers in fully-connected, convolutional, and recurrent neural networks for image classification and language modeling tasks, our new classes exceed the accuracy of existing compression approaches, and on some tasks also outperform general unstructured layers while using more than 20x fewer parameters.

## 1 Introduction

Recent years have seen a surge of interest in structured representations for deep learning, motivated by achieving compression and acceleration while maintaining generalization properties. A popular approach for learning compact models involves constraining the weight matrices to exhibit some form of dense but compressible structure and learning directly over the parameterization of this structure. Examples of structures explored for the weight matrices of deep learning pipelines include low-rank matrices [15, 42], low-distortion projections [49], (block-)circulant matrices [8, 17], Toeplitz-like matrices [34, 45], and constructions derived from Fourier-related transforms [37]. Though they confer significant storage and computation benefits, these constructions tend to underperform general fully-connected layers in deep learning. This raises the question of whether broader classes of structured matrices can achieve superior downstream performance while retaining compression guarantees.

Our approach leverages the **low displacement rank** (LDR) framework (Section 2), which encodes structure through two sparse *displacement operators* and a low-rank residual term [27]. Previous work studying neural networks with LDR weight matrices assumes fixed displacement operators and learns only over the residual [45, 50]. The only case attempted in practice that explicitly employs the LDR framework uses fixed operators encoding shift invariance, producing weight matrices which were found to achieve superior downstream quality than several other compression approaches [45]. Unlike previous work, we consider learning the displacement operators *jointly* with the low-rank residual. Building upon recent progress on structured dense matrix-vector multiplication [14], we introduce a more general class of LDR matrices and develop practical algorithms for using these

---

[*]These authors contributed equally.

matrices in deep learning architectures. We show that the resulting class of matrices subsumes many previously used structured layers, including constructions that did not explicitly use the LDR framework [17, 37]. When compressing weight matrices in fully-connected, convolutional, and recurrent neural networks, we empirically demonstrate improved accuracy over existing approaches. Furthermore, on several tasks our constructions achieve higher accuracy than general unstructured layers while using an order of magnitude fewer parameters.

To shed light on the empirical success of LDR matrices in machine learning, we draw connections to recent work on learning equivariant representations, and hope to motivate further investigations of this link. Notably, many successful previous methods for compression apply classes of structured matrices related to convolutions [8, 17, 45]; while their explicit aim is to accelerate training and reduce memory costs, this constraint implicitly encodes a shift-invariant structure that is well-suited for image and audio data. We observe that the LDR construction enforces a natural notion of approximate equivariance to transformations governed by the displacement operators, suggesting that, in contrast, our approach of learning the operators allows for modeling and learning more general latent structures in data that may not be precisely known in advance.

Despite their increased expressiveness, our new classes retain the storage and computational benefits of conventional structured representations. Our construction provides guaranteed compression (from quadratic to linear parameters) and matrix-vector multiplication algorithms that are quasi-linear in the number of parameters. We additionally provide the first analysis of the sample complexity of learning neural networks with LDR weight matrices, which extends to low-rank, Toeplitz-like and other previously explored fixed classes of LDR matrices. More generally, our analysis applies to structured matrices whose parameters can interact multiplicatively with high degree. We prove that the class of neural networks constructed from these matrices retains VC dimension almost linear in the number of parameters, which implies that LDR matrices with learned displacement operators are still efficiently recoverable from data. This is consistent with our empirical results, which suggest that constraining weight layers to our broad class of LDR matrices can reduce the sample complexity of learning compared to unstructured weights.

We provide a detailed review of previous work and connections to our approach in Appendix B.

**Summary of contributions:**

- We introduce a rich class of LDR matrices where the displacement operators are explicitly learned from data, and provide multiplication algorithms implemented in PyTorch (Section 3).[2]

- We prove that the VC dimension of multi-layer neural networks with LDR weight matrices, which encompasses a broad class of previously explored approaches including the low-rank and Toeplitz-like classes, is quasi-linear in the number of parameters (Section 4).

- We empirically demonstrate that our construction improves downstream quality when compressing weight layers in fully-connected, convolutional, and recurrent neural networks compared to previous compression approaches, and on some tasks can even outperform general unstructured layers (Section 5).

## 2 Background: displacement rank

The generic term *structured matrix* refers to an $m \times n$ matrix that can be represented in much fewer than $mn$ parameters, and admits fast operations such as matrix-vector multiplication. The displacement rank approach represents a structured matrix $\mathbf{M} \in \mathbb{R}^{m \times n}$ through **displacement operators** $(\mathbf{A} \in \mathbb{R}^{m \times m}, \mathbf{B} \in \mathbb{R}^{n \times n})$ defining a linear map $\nabla_{\mathbf{A}, \mathbf{B}} : \mathbf{M} \mapsto \mathbf{AM} - \mathbf{MB}$ on matrices, and a **residual R**, so that if

$$\mathbf{AM} - \mathbf{MB} = \mathbf{R} \tag{1}$$

then $\mathbf{M}$ can be manipulated solely through the compressed representation $(\mathbf{A}, \mathbf{B}, \mathbf{R})$. We assume that $\mathbf{A}$ and $\mathbf{B}$ have disjoint eigenvalues, which guarantees that $\mathbf{M}$ can be recovered from $\mathbf{A}, \mathbf{B}, \mathbf{R}$ (c.f. Theorem 4.3.2, Pan [40]). The rank of $\mathbf{R}$ (also denoted $\nabla_{\mathbf{A}, \mathbf{B}}[\mathbf{M}]$) is called the **displacement rank** of $\mathbf{M}$ w.r.t. $(\mathbf{A}, \mathbf{B})$.[3]

The displacement approach was originally introduced to describe the *Toeplitz-like* matrices, which are not perfectly Toeplitz but still have shift-invariant structure [27]. These matrices have LDR with respect to *shift/cycle* operators. A standard formulation uses $\mathbf{A} = \mathbf{Z}_1, \mathbf{B} = \mathbf{Z}_{-1}$, where $\mathbf{Z}_f = \begin{bmatrix} 0_{1 \times (n-1)} & f \\ \mathbf{I}_{n-1} & 0_{(n-1) \times 1} \end{bmatrix}$ denotes the matrix with 1 on the subdiagonal and $f$ in the top-right corner. The Toeplitz-like matrices have previously been applied in deep learning and kernel approximation, and in several cases have performed significantly better than competing compressed approaches [10, 34, 45]. Figure 1 illustrates the displacement (1) for a Toeplitz matrix, showing how the shift invariant structure of the matrix leads to a residual of rank at most 2.

$$\begin{bmatrix} & & & 1 \\ 1 & & & \\ & \ddots & & \\ & & 1 & \end{bmatrix} \begin{bmatrix} a_0 & a_1 & \cdots & a_{n-1} \\ a_{-1} & a_0 & \ddots & \vdots \\ \vdots & \ddots & \ddots & a_1 \\ a_{-(n-1)} & \cdots & a_{-1} & a_0 \end{bmatrix} - \begin{bmatrix} a_0 & a_1 & \cdots & a_{n-1} \\ a_{-1} & a_0 & \ddots & \vdots \\ \vdots & \ddots & \ddots & a_1 \\ a_{-(n-1)} & \cdots & a_{-1} & a_0 \end{bmatrix} \begin{bmatrix} & & & -1 \\ 1 & & & \\ & \ddots & & \\ & & 1 & \end{bmatrix} = \begin{bmatrix} x & \cdots & y & 2a_0 \\ & & & z \\ & & & \vdots \\ & & & w \end{bmatrix}$$

Figure 1: Displacement equation for a Toeplitz matrix with respect to shift operators $\mathbf{Z}_1, \mathbf{Z}_{-1}$.

A few distinct classes of useful matrices are known to satisfy a displacement property: the classic types are the Toeplitz-, Hankel-, Vandermonde-, and Cauchy-like matrices (Appendix C, Table 5), which are ubiquitous in other disciplines [40]. These classes have fixed operators consisting of diagonal or shift matrices, and LDR properties have traditionally been analyzed in detail only for these special cases. Nonetheless, a few elegant properties hold for generic operators, stating that certain combinations of (and operations on) LDR matrices preserve low displacement rank. We call these *closure properties*, and introduce an additional block closure property that is related to convolutional filter channels (Section 5.2).

We use the notation $\mathcal{D}_{\mathbf{A},\mathbf{B}}^r$ to refer to the matrices of displacement rank $\leq r$ with respect to $(\mathbf{A}, \mathbf{B})$.

**Proposition 1.** *LDR matrices are closed under the following operations:*

   (a) ***Transpose/Inverse*** *If* $\mathbf{M} \in \mathcal{D}_{\mathbf{A},\mathbf{B}}^r$, *then* $\mathbf{M}^T \in \mathcal{D}_{\mathbf{B}^T,\mathbf{A}^T}^r$ *and* $\mathbf{M}^{-1} \in \mathcal{D}_{\mathbf{B},\mathbf{A}}^r$.

   (b) ***Sum*** *If* $\mathbf{M} \in \mathcal{D}_{\mathbf{A},\mathbf{B}}^r$ *and* $\mathbf{N} \in \mathcal{D}_{\mathbf{A},\mathbf{B}}^s$, *then* $\mathbf{M} + \mathbf{N} \in \mathcal{D}_{\mathbf{A},\mathbf{B}}^{r+s}$.

   (c) ***Product*** *If* $\mathbf{M} \in \mathcal{D}_{\mathbf{A},\mathbf{B}}^r$ *and* $\mathbf{N} \in \mathcal{D}_{\mathbf{B},\mathbf{C}}^s$, *then* $\mathbf{MN} \in \mathcal{D}_{\mathbf{A},\mathbf{C}}^{r+s}$.

   (d) ***Block*** *Let* $\mathbf{M}_{ij}$ *satisfy* $\mathbf{M}_{ij} \in \mathcal{D}_{\mathbf{A}_i,\mathbf{B}_j}^r$ *for* $i = 1 \ldots k, j = 1 \ldots \ell$. *Then the* $k \times \ell$ *block matrix* $(\mathbf{M}_{ij})_{ij}$ *has displacement rank* $rk\ell$.

Proposition 1 is proved in Appendix C.

## 3   Learning displacement operators

We consider two classes of new displacement operators. These operators are fixed to be matrices with particular sparsity patterns, where the entries are treated as learnable parameters.

The first operator class consists of **subdiagonal** (plus corner) matrices: $\mathbf{A}_{i+1,i}$, along with the corner $\mathbf{A}_{0,n-1}$, are the only possible non-zero entries. As $\mathbf{Z}_f$ is a special case matching this sparsity pattern, this class is the most direct generalization of Toeplitz-like matrices with learnable operators.

The second class of operators are **tridiagonal** (plus corner) matrices: with the exception of the outer corners $\mathbf{A}_{0,n-1}$ and $\mathbf{A}_{n-1,0}$, $\mathbf{A}_{i,j}$ can only be non-zero if $|i - j| \leq 1$. Figure 2 shows the displacement operators for the Toeplitz-like class and our more general operators. We henceforth let LDR-SD and LDR-TD denote the classes of matrices with low displacement rank with respect to subdiagonal and tridiagonal operators, respectively. Note that LDR-TD contains LDR-SD.

**Expressiveness**   The matrices we introduce can model rich structure and subsume many types of linear transformations used in machine learning. We list some of the structured matrices that have LDR with respect to tridiagonal displacement operators:

**Proposition 2.** *The LDR-TD matrices contain:*

$$
\begin{bmatrix}
0 & & \cdots & 0 & f \\
1 & 0 & & & 0 \\
\vdots & 1 & \ddots & & \vdots \\
0 & \ddots & \ddots & \ddots & \\
0 & 0 & \cdots & 1 & 0
\end{bmatrix}
\quad
\begin{bmatrix}
0 & & \cdots & 0 & x_0 \\
x_1 & 0 & & & 0 \\
\vdots & x_2 & \ddots & & \vdots \\
0 & \ddots & \ddots & \ddots & \\
0 & 0 & \cdots & x_{n-1} & 0
\end{bmatrix}
\quad
\begin{bmatrix}
b_0 & a_0 & \cdots & 0 & s \\
c_0 & b_1 & a_1 & & 0 \\
\vdots & c_1 & \ddots & \ddots & \vdots \\
0 & & \ddots & b_{n-1} & a_{n-2} \\
t & 0 & \cdots & c_{n-2} & b_{n-1}
\end{bmatrix}
$$

Figure 2: The $\mathbf{Z}_f$ operator (left), and our learnable subdiagonal (center) and tridiagonal (right) operators, corresponding to our proposed LDR-SD and LDR-TD classes.

(a) *Toeplitz-like matrices, which themselves include many Toeplitz and circulant variants (including standard convolutional filters - see Section 5.2 and Appendix C, Corollary 1) [8, 17, 45].*

(b) *low-rank matrices.*

(c) *the other classic displacement structures: Hankel-like, Vandermonde-like, and Cauchy-like matrices.*

(d) *orthogonal polynomial transforms, including the Discrete Fourier and Cosine Transforms.*

(e) *combinations and derivatives of these classes via the closure properties (Proposition 1), including structured classes previously used in machine learning such as ACDC [37] and block circulant layers [17].*

These reductions are stated more formally and proved in Appendix C.1. We also include a diagram of the structured matrix classes included by the proposed LDR-TD class in Figure 5 in Appendix C.1.

**Our parameterization**  Given the parameters $\mathbf{A}, \mathbf{B}, \mathbf{R}$, the operation that must ultimately be performed is matrix-vector multiplication by $\mathbf{M} = \nabla_{\mathbf{A},\mathbf{B}}^{-1}[\mathbf{R}]$. Several schemes for explicitly reconstructing $\mathbf{M}$ from its displacement parameters are known for specific cases [41, 44], but do not always apply to our general operators. Instead, we use $\mathbf{A}, \mathbf{B}, \mathbf{R}$ to implicitly construct a slightly different matrix with at most double the displacement rank, which is simpler to work with.

**Proposition 3.** *Let $\mathcal{K}(\mathbf{A}, \mathbf{v})$ denote the $n \times n$ Krylov matrix, defined to have $i$-th column $\mathbf{A}^i \mathbf{v}$. For any vectors $\mathbf{g}_1, \ldots, \mathbf{g}_r, \mathbf{h}_1, \ldots, \mathbf{h}_r \in \mathbb{R}^n$, then the matrix*

$$\sum_{i=1}^{r} \mathcal{K}(\mathbf{A}, \mathbf{g}_i)\mathcal{K}(\mathbf{B}^T, \mathbf{h}_i)^T \tag{2}$$

*has displacement rank at most $2r$ with respect to $\mathbf{A}^{-1}, \mathbf{B}$.*

Thus our representation stores the parameters $\mathbf{A}, \mathbf{B}, \mathbf{G}, \mathbf{H}$, where $\mathbf{A}, \mathbf{B}$ are either subdiagonal or tridiagonal operators (containing $n$ or $3n$ parameters), and $\mathbf{G}, \mathbf{H} \in \mathbb{R}^{n \times r}$. These parameters implicitly define the matrix (2), which is the LDR weight layer we use.

**Algorithms for LDR-SD**  Generic and near-linear time algorithms for matrix-vector multiplication by LDR matrices with even more general operators, including both the LDR-TD and LDR-SD classes, were recently shown to exist [14]. However, complete algorithms were not provided, as they relied on theoretical results such as the transposition principle [6] that only imply the existence of algorithms. Additionally, the recursive polynomial-based algorithms are difficult to implement efficiently. For LDR-SD, we provide explicit and complete near-linear time algorithms for multiplication by (2), as well as substantially simplify them to be useful in practical settings and implementable with standard library operations. We empirically compare the efficiency of our implementation and unstructured matrix-vector multiplication in Figure 8 and Table 14 in Appendix E, showing that LDR-SD accelerates inference by 3.34-46.06x for $n \geq 4096$. We also show results for the low-rank and Toeplitz-like classes, which have a lower computational cost. For LDR-TD, we explicitly construct the $\mathcal{K}(\mathbf{A}, \mathbf{g}_i)$ and $\mathcal{K}(\mathbf{B}^T, \mathbf{h}_i)$ matrices for $i = 1, ..., r$ from Proposition 3 and then apply

the standard $O(n^2)$ matrix-vector multiplication algorithm. Efficient implementations of near-linear time algorithms for LDR-TD are an interesting area of future work.

**Theorem 1.** *Define the simultaneous computation of $k$ Fast Fourier Transforms (FFT), each with size $m$, to be a* batched FFT *with total size $km$.*

*Consider any subdiagonal matrix $\mathbf{A} \in \mathbb{R}^{n \times n}$ and vectors $\mathbf{g}, \mathbf{h} \in \mathbb{R}^n$. Then $\mathcal{K}(\mathbf{A}, \mathbf{g})^T$ or $\mathcal{K}(\mathbf{A}, \mathbf{g})$ can be multiplied by any vector $\mathbf{x}$ by computing $8 \log_2(n)$ batched FFTs, each of total size $2n$. The total number of computations is $O(n \log^2 n)$.*

These algorithms are also automatically differentiable, which we use to compute the gradients when learning. More complete descriptions of these algorithms are presented in Appendix C.

## 4    Theoretical properties of structured matrices

**Complexity of LDR neural networks**    The matrices we use (2) are unusual in that the parameters interact multiplicatively (namely in $\mathbf{A}^i, \mathbf{B}^i$) to implicitly define the actual layer. In contrast, fully-connected layers are linear and other structured layers, such as Fastfood and ACDC [31, 37, 49], are constant degree in their parameters. However, we can prove that this does not significantly change the learnability of our classes:

**Theorem 2.** *Let $\mathcal{F}$ denote the class of neural networks with $L$ LDR layers, $W$ total parameters, and piecewise linear activations. Let $\mathrm{sign}\,\mathcal{F}$ denote the corresponding classification functions, i.e. $\{x \mapsto \mathrm{sign}\, f(x) : f \in \mathcal{F}\}$. The VC dimension of this class is*

$$\mathrm{VCdim}(\mathrm{sign}\,\mathcal{F}) = O(LW \log W).$$

Theorem 2 matches the standard bound for unconstrained weight matrices [4, 24]. This immediately implies a standard PAC-learnable guarantee [47]. Theorem 2 holds for even more general activations and matrices that for example include the broad classes of [14]. The proof is in Appendix D, and we empirically validate the generalization and sample complexity properties of our class in Section 5.3.

**Displacement rank and equivariance**    We observe that displacement rank is related to a line of work outside the resource-constrained learning community, specifically on building **equivariant** (also called covariant in some contexts [5, 35]) feature representations that transform in predictable ways when the input is transformed. An equivariant feature map $\Phi$ satisfies

$$\Phi(B(x)) = A(\Phi(x)) \tag{3}$$

for transformations $A, B$ (invariance is the special case when $A$ is the identity) [16, 33, 43]. This means that perturbing the input by a transformation $B$ before passing through the map $\Phi$ is equivalent to first finding the features $\Phi$ then transforming by $A$.

Intuitively, LDR matrices are a suitable choice for modeling *approximately equivariant* linear maps, since the residual $\mathbf{A}\Phi - \Phi\mathbf{B}$ of (3) has low complexity. Furthermore, approximately equivariant maps should retain the compositional properties of equivariance, which LDR satisfies via Proposition 1. For example, Proposition 1(c) formalizes the notion that the composition of two approximately equivariant maps is still approximately equivariant. Using this intuition, the displacement representation (1) of a matrix decomposes into two parts: the operators $\mathbf{A}, \mathbf{B}$ define transformations to which the model is approximately equivariant, and the low complexity residual $\mathbf{R}$ controls standard model capacity.

Equivariance has been used in several ways in the context of machine learning. One formulation, used for example to model ego-motions, supposes that (3) holds only approximately, and uses a fixed transformation $B$ along with data for (3) to learn an appropriate $A$ [1, 33]. Another line of work uses the representation theory formalization of equivariant maps [12, 28]. We describe this formulation in more detail and show how LDR satisfies this definition as well in Appendix C.3, Proposition 7. In contrast to previous settings, which fix one or both of $A, B$, our formulation stipulates that $\Phi$ can be uniquely determined from $A$, $B$, and learns the latter as part of an end-to-end model. In Section 5.4 we include a visual example of latent structure that our displacement operators learn, where they recover centering information about objects from a 2D image dataset.

# 5 Empirical evaluation

**Overview**  In Section 5.1 we consider a standard setting of compressing a single hidden layer (SHL) neural network and the fully-connected (FC) layer of a CNN for image classification tasks. Following previous work [7, 45], we test on two challenging MNIST variants [30], and include two additional datasets with more realistic objects (CIFAR-10 [29] and NORB [32]). Since SHL models take a single channel as input, we converted CIFAR-10 to grayscale for this task. Our classes and the structured baselines are tested across different parameter budgets in order to show tradeoffs between compression and accuracy. As shown in Table 1, in the SHL model, our methods consistently have higher test accuracy than baselines for compressed training and inference, by 3.14, 2.70, 3.55, and 3.37 accuracy points on MNIST-bg-rot, MNIST-noise, CIFAR-10, and NORB respectively. In the CNN model, as shown in Table 1 in Appendix E, we found improvements of 5.56, 0.95, and 1.98 accuracy points over baselines on MNIST-bg-rot, MNIST-noise, and NORB respectively. Additionally, to explore whether learning the displacement operators can facilitate adaptation to other domains, we replace the input-hidden weights in an LSTM for a language modeling task, and show improvements of 0.81-30.47 perplexity points compared to baselines at several parameter budgets.

In addition to experiments on replacing fully-connected layers, in Section 5.2 we also replace the convolutional layer of a simple CNN while preserving performance within 1.05 accuracy points on CIFAR-10. In Section 5.3, we consider the effect of a higher parameter budget. By increasing the rank to just 16, the LDR-SD class meets or exceeds the accuracy of the unstructured FC layer in all datasets we tested on, for both SHL and CNN.[4] Appendix F includes more experimental details and protocols. Our PyTorch code is publicly available at `github.com/HazyResearch/structured-nets`.

## 5.1 Compressing fully-connected layers

**Image classification**  Sindhwani et al. [45] showed that for a fixed parameter budget, the Toeplitz-like class significantly outperforms several other compression approaches, including Random Edge Removal [11], Low Rank Decomposition [15], Dark Knowledge [25], HashedNets [7], and HashedNets with Dark Knowledge. Following previous experimental settings [7, 45], Table 1 compares our proposed classes to several baselines using dense structured matrices to compress the hidden layer of a single hidden layer neural network. In addition to Toeplitz-like, we implement and compare to other classic LDR types, Hankel-like and Vandermonde-like, which were previously indicated as an unexplored possibility [45, 50]. We also show results when compressing the FC layer of a 7-layer CNN based on LeNet in Appendix E, Table 7. In Appendix E, we show comparisons to additional baselines at multiple budgets, including network pruning [23] and a baseline used in [7], in which the number of hidden units is adjusted to meet the parameter budget.

At rank one (the most compressed setting), our classes with learned operators achieve higher accuracy than the fixed operator classes, and on the MNIST-bg-rot, MNIST-noise, and NORB datasets even improve on FC layers of the same dimensions, by 1.73, 13.30, and 2.92 accuracy points respectively on the SHL task, as shown in Table 1. On the CNN task, our classes improve upon unstructured fully-connected layers by 0.85 and 2.25 accuracy points on the MNIST-bg-rot and MNIST-noise datasets (shown in Table 7 in Appendix E). As noted above, at higher ranks our classes meet or improve upon the accuracy of FC layers on all datasets in both the SHL and CNN architectures.

Additionally, in Figure 3 we evaluate the performance of LDR-SD at higher ranks. Note that the ratio of parameters between LDR-SD and the Toeplitz-like or low-rank is $\frac{r+1}{r}$, which becomes negligible at higher ranks. Figure 3 shows that at just rank 16, the LDR-SD class meets or exceeds the performance of the FC layer on all four datasets, by 5.87, 15.05, 0.74, and 6.86 accuracy points on MNIST-bg-rot, MNIST-noise, CIFAR-10, and NORB respectively, while still maintaining at least 20x fewer parameters.

Of particular note is the poor performance of low-rank matrices. As mentioned in Section 2, every fixed-operator class has the same parameterization (a low-rank matrix). We hypothesize that the main contribution to their marked performance difference is the effect of the learned displacement operator modeling latent invariances in the data, and that the improvement in the displacement

Table 1: Test accuracy when replacing the hidden layer with structured classes. Where applicable, rank ($r$) is in parentheses, and the number of parameters in the architecture is in italics below each method. Comparisons to previously unexplored classic LDR types as well as additional structured baselines are included, with the ranks adjusted to match the parameter count of LDR-TD where possible. The Fastfood [49] and Circulant [8] methods do not have rank parameters, and the parameter count for these methods cannot be exactly controlled. Additional results when replacing the FC layer of a CNN are in Appendix E. Details for all experiments are in Appendix F.

| Method | MNIST-bg-rot | MNIST-noise | CIFAR-10 | NORB |
|---|---|---|---|---|
| Unstructured | 44.08 | 65.15 | 46.03 | 59.83 |
| | *622506* | *622506* | *1058826* | *1054726* |
| LDR-TD ($r = 1$) | **45.81** | **78.45** | **45.33** | **62.75** |
| | *14122* | *14122* | *18442* | *14342* |
| Toeplitz-like [45] ($r = 4$) | 42.67 | 75.75 | 41.78 | 59.38 |
| | *14122* | *14122* | *18442* | *14342* |
| Hankel-like ($r = 4$) | 42.23 | 73.65 | 41.40 | 60.09 |
| | *14122* | *14122* | *18442* | *14342* |
| Vandermonde-like ($r = 4$) | 37.14 | 59.80 | 33.93 | 48.98 |
| | *14122* | *14122* | *18442* | *14342* |
| Low-rank [15] ($r = 4$) | 35.67 | 52.25 | 32.28 | 43.66 |
| | *14122* | *14122* | *18442* | *14342* |
| Fastfood [49] | 38.13 | 63.55 | 39.64 | 59.02 |
| | *10202* | *10202* | *13322* | *9222* |
| Circulant [8] | 34.46 | 65.35 | 34.28 | 46.45 |
| | *8634* | *8634* | *11274* | *7174* |

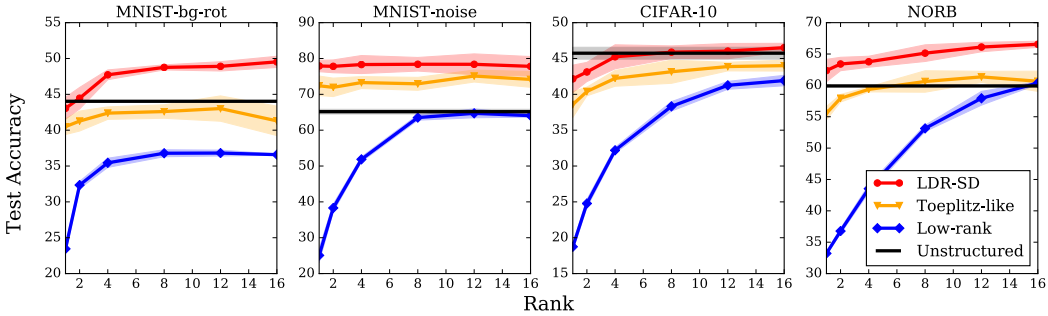

Figure 3: Test accuracy vs. rank for unstructured, LDR-SD, Toeplitz-like, low-rank classes. On each dataset, LDR-SD meets or exceeds the accuracy of the unstructured FC baseline at higher ranks. At rank 16, the compression ratio of an LDR-SD layer compared to the unstructured layer ranges from 23 to 30. Shaded regions represent two standard deviations from the mean, computed over five trials with randomly initialized weights.

rank classes—from low-rank to Toeplitz-like to our learned operators—comes from more accurate representations of these invariances. As shown in Figure 3, broadening the operator class (from Toeplitz-like at $r = 1$ to LDR-SD at $r = 1$) is consistently a more effective use of parameters than increasing the displacement rank (from Toeplitz-like at $r = 1$ to $r = 2$). Note that LDR-SD ($r = 1$) and Toeplitz-like ($r = 2$) have the same parameter count.

For the rest of our experiments outside Section 5.1 we use the algorithms in Appendix C specifically for LDR-SD matrices, and focus on further evaluation of this class on more expensive models.

**Language modeling**    Here, we replace the input-hidden weights in a single layer long short-term memory network (LSTM) for a language modeling task. We evaluate on the WikiText-2 dataset, consisting of 2M training tokens and a vocabulary size of 33K [36]. We compare to Toeplitz-like and low-rank baselines, both previously investigated for compressing recurrent nets [34]. As shown in Table 2, LDR-SD improves upon the baselines for each budget tested. Though our class does

not outperform the unstructured model, we did find that it achieves a significantly lower perplexity than the fixed Toeplitz-like class (by 19.94-42.92 perplexity points), suggesting that learning the displacement operator can help adapt to different domains.

Table 2: Test perplexity when replacing input-hidden matrices of an LSTM with structured classes on WikiText-2. An unconstrained layer, with 65536 parameters, has perplexity 117.74. Parameter budgets correspond to ranks 1,2,4,8,16,24 for LDR-SD. Lower is better.

| Num. Parameters | LDR-SD | Toeplitz-like | Low-rank |
|---|---|---|---|
| 2048 | **166.97** | 186.91 | 205.72 |
| 3072 | **154.51** | 177.60 | 179.46 |
| 5120 | **141.91** | 178.07 | 172.38 |
| 9216 | **143.60** | 186.52 | 144.41 |
| 17408 | **132.43** | 162.58 | 135.65 |
| 25600 | **129.46** | 155.73 | 133.37 |

## 5.2 Replacing convolutional layers

Convolutional layers of CNNs are a prominent example of equivariant feature maps.[5] It has been noted that convolutions are a subcase of Toeplitz-like matrices with a particular sparsity pattern[6] [8, 45]. As channels are simply block matrices[7], the block closure property implies that multi-channel convolutional filters are simply a Toeplitz-like matrix of higher rank (see Appendix C, Corollary 1). In light of the interpretation of LDR of an approximately equivariant linear map (as discussed in Section 4), we investigate whether replacing convolutional layers with more general representations can recover similar performance, without needing the hand-crafted sparsity pattern.

Briefly, we test the simplest multi-channel CNN model on the CIFAR-10 dataset, consisting of one layer of convolutional channels (3 in/out channels), followed by a FC layer, followed by the softmax layer. The final accuracies are listed in Table 3. The most striking result is for the simple architecture consisting of two layers of a single structured matrix. This comes within 1.05 accuracy points of the highly specialized architecture consisting of convolutional channels + pooling + FC layer, while using fewer layers, hidden units, and parameters. The full details are in Appendix F.

Table 3: Replacing a five-layer CNN consisting of convolutional channels, max pooling, and FC layers with two generic LDR matrices results in only slight test accuracy decrease while containing fewer layers, hidden units, and parameters. Rank $(r)$ is in parentheses.

| First hidden layer(s) | Last hidden layer | Hidden units | Parameters | Test Acc. |
|---|---|---|---|---|
| 3 Convolutional Channels (CC) | FC | 3072, 512 | 1573089 | 54.59 |
| 3CC + Max Pool | FC | 3072, 768, 512 | 393441 | 55.14 |
| 4CC + Max Pool | FC | 4096, 1024, 512 | 524588 | **60.05** |
| Toeplitz-like $(r = 16)$ channels | Toeplitz-like $(r = 16)$ | 3072, 512 | 393216 | 57.29 |
| LDR-SD $(r = 16)$ channels | LDR-SD $(r = 16)$ | 3072, 512 | 417792 | 59.36 |
| Toeplitz-like $(r = 48)$ matrix | Toeplitz-like $(r = 16)$ | 3072, 512 | 393216 | 55.29 |
| LDR-SD $(r = 48)$ matrix | LDR-SD $(r = 16)$ | 3072, 512 | 405504 | **59.00** |

## 5.3 Generalization and sample complexity

Theorem 2 states that the theoretical sample complexity of neural networks with structured weight matrices scales almost linearly in the total number of parameters, matching the results for networks with fully-connected layers [4, 24]. As LDR matrices have far fewer parameters, the VC dimension

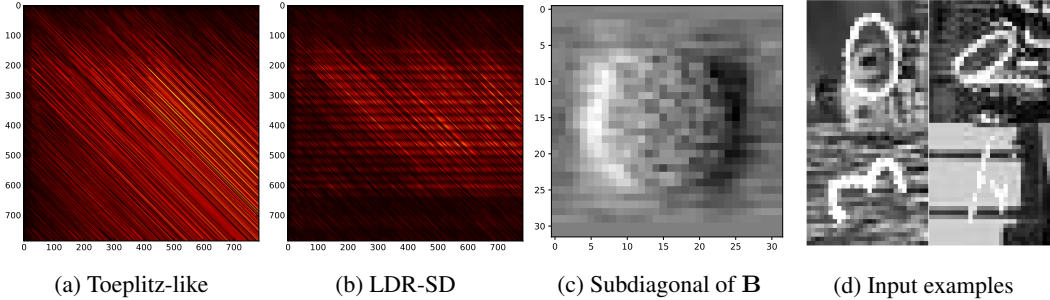

| (a) Toeplitz-like | (b) LDR-SD | (c) Subdiagonal of $\mathbf{B}$ | (d) Input examples |

Figure 4: The learned weight matrices (a,b) of models trained on MNIST-bg-rot. Unlike the Toeplitz-like matrix, the LDR-SD matrix displays grid-like periodicity corresponding to the 2D input. Figure (c) shows the values of the subdiagonal of $\mathbf{B}$, reshaped as an image. The size and location of the circle roughly corresponds to the location of objects of interest in the 2D inputs. A similar centering phenomenon was found on the NORB dataset, shown in Figure 6 in Appendix E.

bound for LDR networks are correspondingly lower than that of general unstructured networks. Though the VC dimension bounds are sufficient but not necessary for learnability, one might still expect to be able to learn over compressed networks with fewer samples than over unstructured networks. We empirically investigate this result using the same experimental setting as Table 1 and Figure 3. As shown in Table 12 (Appendix E), the structured classes consistently have lower generalization error (measured by the difference between training and test error) than the unstructured baseline.

**Reducing sample complexity** We investigate whether LDR models with learned displacement operators require fewer samples to achieve the same test error, compared to unstructured weights, in both the single hidden layer and CNN architectures. Tables 10 and 11 in Appendix E show our results. In the single hidden layer architecture, when using only 25% of the training data the LDR-TD class exceeds the performance of an unstructured model trained on the full MNIST-noise dataset. On the CNN model, only 50% of the training data is sufficient for the LDR-TD to exceed the performance of an unstructured layer trained on the full dataset.

## 5.4 Visualizing learned weights

Finally, we examine the actual structures that our models learn. Figure 4(a,b) shows the heat map of the weight matrix $\mathbf{W} \in \mathbb{R}^{784 \times 784}$ for the Toeplitz-like and LDR-SD classes, trained on MNIST-bg-rot with a single hidden layer model. As is convention, the input is flattened to a vector in $\mathbb{R}^{784}$. The Toeplitz-like class is unable to determine that the input is actually a $28 \times 28$ image instead of a vector. In contrast, LDR-SD class is able to pick up regularity in the input, as the weight matrix displays grid-like periodicity of size 28.

Figure 4(c) reveals why the weight matrix displays this pattern. The equivariance interpretation (Section 4) predicts that $\mathbf{B}$ should encode a meaningful transformation of the inputs. The entries of the learned subdiagonal are in fact recovering a latent invariant of the 2D domain: when visualized as an image, the pixel intensities correspond to how the inputs are centered in the dataset (Figure 4(d)). Figure 6 in Appendix E shows a similar figure for the NORB dataset, which has smaller objects, and we found that the subdiagonal learns a correspondingly smaller circle.

## 6 Conclusion

We generalize the class of low displacement rank matrices explored in machine learning by considering classes of LDR matrices with displacement operators that can be learned from data. We show these matrices can improve performance on downstream tasks compared to compression baselines and, on some tasks, general unstructured weight layers. We hope this work inspires additional ways of using structure to achieve both more compact and higher quality representations, especially for deep learning models, which are commonly acknowledged to be overparameterized.

## Footnotes

[2]Our code is available at `https://github.com/HazyResearch/structured-nets`.

[3]Throughout this paper, we use square matrices for simplicity, but LDR is well-defined for rectangular.

[4]In addition to the results reported in Table 1, Figure 3 and Table 7 in Appendix E, we also found that at rank 16 the LDR-SD class on the CNN architecture achieved test accuracies of 68.48% and 75.45% on CIFAR-10 and NORB respectively.

[5]Convolutions are designed to be shift equivariant, i.e. shifting the input is equivalent to shifting the output.

[6]E.g. a $3 \times 3$ convolutional filter on an $n \times n$ matrix has a Toeplitz weight matrix supported on diagonals $-1, 0, 1, n-1, n, n+1, 2n-1, \ldots$.

[7]A layer consisting of $k$ in-channels and $\ell$ out-channels, each of which is connected by a weight matrix of class $\mathcal{C}$, is the same as a $k \times \ell$ block matrix.
