[Supplementary Material]

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

**Acknowledgments**

We thank Taco Cohen, Jared Dunnmon, Braden Hancock, Tatsunori Hashimoto, Fred Sala, Virginia Smith, James Thomas, Mary Wootters, Paroma Varma, and Jian Zhang for helpful discussions and feedback.

We gratefully acknowledge the support of DARPA under Nos. FA87501720095 (D3M) and FA86501827865 (SDH), NIH under No. N000141712266 (Mobilize), NSF under Nos. CCF1763315 (Beyond Sparsity) and CCF1563078 (Volume to Velocity), ONR under No. N000141712266 (Unifying Weak Supervision), the Moore Foundation, NXP, Xilinx, LETI-CEA, Intel, Google, NEC, Toshiba, TSMC, ARM, Hitachi, BASF, Accenture, Ericsson, Qualcomm, Analog Devices, the Okawa Foundation, and American Family Insurance, and members of the Stanford DAWN project: Intel, Microsoft, Teradata, Facebook, Google, Ant Financial, NEC, SAP, and VMWare. The U.S. Government is authorized to reproduce and distribute reprints for Governmental purposes notwithstanding any copyright notation thereon. Any opinions, findings, and conclusions or recommendations expressed in this material are those of the authors and do not necessarily reflect the views, policies, or endorsements, either expressed or implied, of DARPA, NIH, ONR, or the U.S. Government.

## Footnotes

[2]Our code is available at `https://github.com/HazyResearch/structured-nets`.

[3]Throughout this paper, we use square matrices for simplicity, but LDR is well-defined for rectangular.

[4]In addition to the results reported in Table 1, Figure 3 and Table 7 in Appendix E, we also found that at rank 16 the LDR-SD class on the CNN architecture achieved test accuracies of 68.48% and 75.45% on CIFAR-10 and NORB respectively.

[5]Convolutions are designed to be shift equivariant, i.e. shifting the input is equivalent to shifting the output.

[6]E.g. a $3 \times 3$ convolutional filter on an $n \times n$ matrix has a Toeplitz weight matrix supported on diagonals $-1, 0, 1, n - 1, n, n + 1, 2n - 1, \dots$.

[7]A layer consisting of $k$ in-channels and $\ell$ out-channels, each of which is connected by a weight matrix of class $\mathcal{C}$, is the same as a $k \times \ell$ block matrix.

[8]Shifting the input to a convolutional feature map is the same as shifting the output.

[9]The convolutions are padded to ensure their input and output dimensions are equal.

[10]Code available at `https://github.com/pytorch/examples/tree/master/word_language_model`.

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

# A  Symbols and abbreviations

Table 4: Symbols and abbreviations used in this paper.

| Symbol | Used For |
|---|---|
| LDR | low displacement rank |
| LDR-SD | matrices with low displacement rank with respect to subdiagonal operators |
| LDR-TD | matrices with low displacement rank with respect to tridiagonal operators |
| $(\mathbf{A}, \mathbf{B})$ | displacement operators |
| $\nabla_{\mathbf{A},\mathbf{B}}[\mathbf{M}]$ | Sylvester displacement, $\mathbf{A}\mathbf{M} - \mathbf{M}\mathbf{B}$ |
| $r$ | (displacement) rank |
| $(\mathbf{G}, \mathbf{H})$ | parameters which define the rank $r$ residual matrix $\mathbf{G}\mathbf{H}^T$, where $\mathbf{G}, \mathbf{H} \in \mathbb{R}^{n \times r}$ |
| $\mathbf{Z_f}$ | unit-f-circulant matrix, defined as $\mathbf{Z_f} = [\mathbf{e_2}, \mathbf{e_3}, ..., \mathbf{e_n}, f\mathbf{e_1}]$ |
| $\mathcal{K}(\mathbf{A}, \mathbf{v})$ | Krylov matrix, with $i^{th}$ column $\mathbf{A}^i \mathbf{v}$ |
| $\mathcal{D}_{\mathbf{A},\mathbf{B}}^r$ | matrices of displacement rank $\leq r$ with respect to $(\mathbf{A}, \mathbf{B})$ |
| $\Phi$ | feature map |
| CC | convolutional channels |
| FC | fully-connected |

# B  Related work

Our study of the potential for structured matrices for compressing deep learning pipelines was motivated by exciting work along these lines from Sindhwani et al. [45], the first to suggest the use of low displacement rank (LDR) matrices in deep learning. They specifically explored applications of the Toeplitz-like class, and empirically show that this class is competitive against many other baselines for compressing neural networks on image and speech domains. Toeplitz-like matrices were similarly found to be effective at compressing RNN and LSTM architectures on a voice search task [34]. Another special case of LDR matrices are the circulant (or block-circulant) matrices, which have also been used for compressing CNNs [8]; more recently, these have also been further developed and shown to achieve state-of-the-art results on FPGA and ASIC platforms [17]. Earlier works on compressing deep learning pipelines investigated the use of low-rank matrices [15, 42]—perhaps the most canonical type of dense structured matrix—which are also encompassed by our framework, as shown in Proposition 2. Outside of deep learning, Choromanski and Sindhwani [10] examined a structured matrix class that includes Toeplitz-like, circulant, and Hankel matrices (which are all LDR matrices) in the context of kernel approximation.

On the theoretical side, Zhao et al. [50] study properties of neural networks with LDR weight matrices, proving results including a universal approximation property and error bounds. However, they retain the standard paradigm of fixing the displacement operators and varying the low-rank portion. Another natural theoretical question that arises with these models is whether the resulting hypothesis class is still efficiently learnable, especially when learning the structured class (as opposed to these previous fixed classes). Recently, Oymak [38] proved a Rademacher complexity bound for one layer neural networks with low-rank weight matrices. To the best of our knowledge, Theorem 2 provides the first sample complexity bounds for neural networks with a broad class of structured weight matrices including low-rank, our LDR classes, and other general structured matrices [14].

In Section 3 we suggest that the LDR representation enforces a natural notion of approximate equivariance and satisfies closure properties that one would expect of equivariant representations. The study of equivariant feature maps is of broad interest for constructing more effective representations when known symmetries exist in underlying data. Equivariant linear maps have long been used in algebraic signal processing to derive efficient transform algorithms [18, 19]. The fact that convolutional networks induce equivariant representations, and the importance of this effect on sample complexity and generalization, has been well-analyzed [2, 12, 21, 46]. Building upon the observation that convolutional filters are simply linear maps constructed to be translation equivariant[8], exciting recent progress has been made on crafting representations invariant to more complex symmetries such as the spherical rotation group [13] and egomotions [1]. Generally, however, underlying assumptions

are made about the domain and invariances present in order to construct feature maps for each application. A few works have explored the possibility of learning invariances automatically from data, and design deep architectures that are in principle capable of modeling and learning more general symmetries [20, 26, 39].

## C  Properties of displacement rank

Displacement rank has traditionally been used to describe the Toeplitz-like, Hankel-like, Vandermonde-like, and Cauchy-like matrices, which are ubiquitous in disciplines such as engineering, coding theory, and computer algebra. Their associated displacement representations are shown in Table 5.

Table 5: Traditional classes of structured matrices analyzed with displacement rank. In the Vandermonde and Cauchy cases, the displacement operators are parameterized by $v \in \mathbb{R}^n$ and $s, t \in \mathbb{R}^n$ respectively.

| Structured Matrix $\mathbf{M}$ | $\mathbf{A}$ | $\mathbf{B}$ | Displacement Rank $r$ |
|---|---|---|---|
| Toeplitz | $\mathbf{Z}_1$ | $\mathbf{Z}_{-1}$ | $\leq 2$ |
| Hankel | $\mathbf{Z}_1$ | $\mathbf{Z}_0^T$ | $\leq 2$ |
| Vandermonde | $\mathrm{diag}(v)$ | $\mathbf{Z}_0$ | $\leq 1$ |
| Cauchy | $\mathrm{diag}(s)$ | $\mathrm{diag}(t)$ | $\leq 1$ |

*Proof of Proposition 1.* The following identities are easily verified:

**Transpose**
$$\nabla_{\mathbf{B}^T, \mathbf{A}^T} \mathbf{M}^T = -\left(\nabla_{\mathbf{A},\mathbf{B}} \mathbf{M}\right)^T$$

**Inverse**
$$\nabla_{\mathbf{B},\mathbf{A}} \mathbf{M}^{-1} = -\mathbf{M}^{-1}\left(\nabla_{\mathbf{A},\mathbf{B}} \mathbf{M}\right)\mathbf{M}^{-1}$$

**Sum**
$$\nabla_{\mathbf{A},\mathbf{B}}(\mathbf{M} + \mathbf{N}) = \nabla_{\mathbf{A},\mathbf{B}}\mathbf{M} + \nabla_{\mathbf{A},\mathbf{B}}\mathbf{N}$$

**Product**
$$\nabla_{\mathbf{A},\mathbf{C}}\mathbf{M}\mathbf{N} = (\nabla_{\mathbf{A},\mathbf{B}}\mathbf{M})\mathbf{N} + \mathbf{M}\left(\nabla_{\mathbf{B},\mathbf{C}}\mathbf{N}\right)$$

**Block**  The remainder
$$\mathrm{diag}(\mathbf{A}_1, \dots, \mathbf{A}_k)\mathbf{M} - \mathbf{M}\,\mathrm{diag}(\mathbf{B}_1, \dots, \mathbf{B}_\ell)$$
is the block matrix
$$(\nabla_{\mathbf{A}_i, \mathbf{B}_j}\mathbf{M}_{ij})_{1 \leq i \leq k, 1 \leq j \leq \ell}.$$
This is the sum of $k\ell$ matrices of rank $r$ and thus has rank $rk\ell$.

$\square$

**Corollary 1.** *A $k \times \ell$ block matrix $\mathbf{M}$, where each block is a Toeplitz-like matrix of displacement rank $r$, is Toeplitz-like with displacement rank $rk\ell + 2k + 2\ell$.*

*Proof.* Apply Proposition (d) where each $\mathbf{A}_k, \mathbf{B}_k$ has the form $\mathbf{Z}_f$. Let $\mathbf{A} = \mathrm{diag}(\mathbf{A}_1, \dots, \mathbf{A}_k)$ and $\mathbf{B} = \mathrm{diag}(\mathbf{B}_1, \dots, \mathbf{B}_\ell)$. Note that $\mathbf{A}$ and $\mathbf{Z}_1$ (of the same size as $\mathbf{A}$) differ only in $2k$ entries, and similarly $\mathbf{B}$ and $\mathbf{Z}_{-1}$ differ in $2\ell$ entries. Since an $s$-sparse matrix also has rank at most $s$,
$$\mathbf{Z}_1\mathbf{M} - \mathbf{M}\mathbf{Z}_{-1} = \mathbf{A}\mathbf{M} - \mathbf{M}\mathbf{B} + (\mathbf{Z}_1 - \mathbf{A})\mathbf{M} - \mathbf{M}(\mathbf{Z}_{-1} - \mathbf{B})$$
has rank at most $rk\ell + 2k + 2\ell$. $\square$

*Proof of Proposition 3.* First consider the rank one case, $\mathbf{R} = \mathbf{g}\mathbf{h}^T$. It is easy to check that $\nabla_{\mathbf{A}^{-1}, \mathbf{Z}^T}\mathcal{K}(\mathbf{A}, \mathbf{g})$ will only be non-empty in the first column, hence $\mathcal{K}(\mathbf{A}, \mathbf{g}) \in \mathcal{D}^1_{\mathbf{A}^{-1}, \mathbf{Z}^T}$. Similarly, $\mathcal{K}(\mathbf{B}^T, \mathbf{h}) \in \mathcal{D}^1_{\mathbf{B}^T, \mathbf{Z}}$ and Proposition 1(a) implies $\mathcal{K}(\mathbf{B}^T, \mathbf{h})^T \in \mathcal{D}^1_{\mathbf{Z}^T, \mathbf{B}}$. Then Theorem 1(c) implies that $\mathcal{K}(\mathbf{A}, \mathbf{g})\mathcal{K}(\mathbf{B}, \mathbf{h})^T \in \mathcal{D}^2_{\mathbf{A}, \mathbf{B}}$. The rank $r$ case follows directly from Theorem 1(b). $\square$

## C.1 Expressiveness

Expanding on the claim in Section 3, we formally show that these structured matrices are contained in the tridiagonal (plus corners) LDR class. This includes several types previously used in similar works.

Figure 5: Our proposed LDR-TD structured matrix class contains a number of other classes including Toeplitz-like [45] (and other classic displacement types, such as Hankel-like, Vandermonde-like, and Cauchy-like), low-rank [15], circulant [8], standard convolutional filters, and orthogonal polynomial transforms, including the Discrete Fourier and Cosine Transforms. Captions for each class show storage cost and operation count for matrix-vector multiplication.

**Classic displacement rank**   The Toeplitz-like, Hankel-like, Vandermonde-like, and Cauchy-like matrices are defined as having LDR with respect to $\mathbf{A}, \mathbf{B} \in \{\mathbf{Z}_f, \mathbf{Z}_f^T, \mathbb{D}\}$ where $\mathbb{D}$ is the set of diagonal matrices [40]. (For example, [45] defines the Toeplitz-like matrices as $(\mathbf{A}, \mathbf{B}) = (\mathbf{Z}_1, \mathbf{Z}_{-1})$.) All of these operator choices are only non-zero along the three main diagonals or opposite corners, and hence these classic displacement types belong to the LDR-TD class.

**Low-rank**   A rank $r$ matrix $R$ trivially has displacement rank $r$ with respect to $(\mathbf{A}, \mathbf{B}) = (\mathbf{I}, \mathbf{0})$. It also has displacement rank $r$ with respect to $(\mathbf{A}, \mathbf{B}) = (\mathbf{Z}_1, \mathbf{0})$, since $\mathbf{Z}_1$ is full rank (it is a permutation matrix) and so $\mathrm{rank}(Z_1 R) = \mathrm{rank}(R) = r$. Thus low-rank matrices are contained in both the LDR-TD and LDR-SD classes.

**Orthogonal polynomial transforms**   The **polynomial transform** matrix $\mathbf{M}$ with respect to polynomials $(p_0(X), \ldots, p_{m-1}(X))$ and nodes $(\lambda_0, \ldots, \lambda_{n-1})$ is defined by $\mathbf{M}_{ij} = p_i(\lambda_j)$. When the $p_i(X)$ are a family of orthogonal polynomials, it is called an **orthogonal polynomial transform**.

**Proposition 4.** *Orthogonal polynomial transforms have displacement rank* 1 *with respect to tridiagonal operators.*

*Proof.* Every orthogonal polynomial family satisfies a three-term recurrence

$$p_{i+1}(X) = (a_i X + b_i) p_i(X) + c_i p_{i-1}(X) \tag{4}$$

where $a_i > 0$ [9]. Let $\mathbf{M}$ be an orthogonal polynomial transform with respect to polynomials $(p_i(X))_{0 \le i < m}$ and nodes $(\lambda_j)_{0 \le j < n}$. Define the tridiagonal and diagonal matrix

$$\mathbf{A} = \begin{bmatrix} -\frac{b_0}{a_0} & \frac{1}{a_0} & 0 & \dots & 0 & 0 \\ -\frac{c_1}{a_1} & -\frac{b_1}{a_1} & \frac{1}{a_1} & \dots & 0 & 0 \\ 0 & -\frac{c_1}{a_1} & -\frac{b_1}{a_1} & \dots & 0 & 0 \\ \vdots & \vdots & \vdots & \ddots & \vdots & \vdots \\ 0 & 0 & 0 & \dots & -\frac{b_{m-2}}{a_{m-2}} & \frac{1}{a_{m-2}} \\ 0 & 0 & 0 & \dots & -\frac{c_{m-1}}{a_{m-1}} & -\frac{b_{m-1}}{a_{m-1}} \end{bmatrix}$$

$$\mathbf{B} = \operatorname{diag}(\lambda_0, \dots, \lambda_{n-1}).$$

For any $i \in \{0, \dots, m-2\}$ and any $j$, consider entry $ij$ of $\mathbf{AM} - \mathbf{MB}$. This is

$$\frac{1}{a_i} \left[ -c_i p_{i-1}(\lambda_j) - b_i p_i(\lambda_j) + p_{i+1}(\lambda_j) - \lambda_j p_i(\lambda_j) \right]$$

which is 0 by plugging $\lambda_j$ into (4).

Thus $\nabla_{\mathbf{A},\mathbf{B}} \mathbf{M}$ can only non-zero in the last row, so $\mathbf{M} \in \mathcal{D}^1_{\mathbf{A},\mathbf{B}}$. $\qquad \square$

**Fourier-like transforms**    Orthogonal polynomial transforms include many special cases. We single out the Discrete Fourier Transform (DFT) and Discrete Cosine Transform (DCT) for their ubiquity.

The $N \times N$ DFT and DCT (type II) are defined as matrix multiplication by the matrices

$$\mathbf{F} = \left( e^{-2\pi \frac{ij}{N}} \right)_{ij}$$

$$\mathbf{C} = \left( \cos\left[ \frac{\pi}{N} i(j + 1/2) \right] \right)_{ij}$$

respectively.

The former is a special type of Vandermonde matrix, which were already shown to be in LDR-TD. Also note that Vandermonde matrices $(\lambda_j^i)_{ij}$ are themselves orthogonal polynomial transforms with $p_i(X) = X^i$.

The latter can be written as

$$\left( T_i \left( \cos\left[ \frac{\pi}{N} (j + \frac{1}{2}) \right] \right) \right)_{ij},$$

where $T_i$ are the **Chebyshev polynomials** (of the first kind) defined such that

$$T_n(X) = \cos(n \arccos x).$$

Thus this is an orthogonal polynomial transform with respect to the Chebyshev polynomials.

**Other constructions**    From these basic building blocks, interesting constructions belonging to LDR-TD can be found via the closure properties. For example, several types of structured layers inspired by convolutions, including Toeplitz [45], circulant [8] and block-circulant [17] matrices, are special instances of Toeplitz-like matrices. We also point out a more sophisticated layer [37] in the tridiagonal LDR class, which requires more deliberate use of Proposition 1 to show.

**Proposition 5.** *The* $\mathbf{ACDC}^{-1}$ *layer, where* $\mathbf{A}, \mathbf{D}$ *are diagonal matrices and* $\mathbf{C}$ *is the Discrete Cosine Transform [37], has displacement rank 2 with respect to tridiagonal operators.*

*Proof.* Let $\mathbf{T}, \Lambda$ be the tridiagonal and diagonal matrix such that $\mathbf{C} \in \mathcal{D}^1_{\mathbf{T},\Lambda}$. Define $\mathbf{S} = \mathbf{ATA}^{-1}$, which is also tridiagonal. Note that $\mathbf{A} \in \mathcal{D}^0_{\mathbf{S},\mathbf{T}}$ by construction. Also note that $\mathbf{D} \in \mathcal{D}^0_{\Lambda,\Lambda}$ since $\Lambda$ is diagonal. An application of the inverse closure rule yields $\mathbf{C} \in \mathcal{D}^1_{\Lambda,\mathbf{T}}$. Finally, the product closure property implies that

$$\mathbf{ACDC}^{-1} \in \mathcal{D}^2_{\mathbf{S},\mathbf{T}}.$$

$\qquad \square$

## C.2 Algorithm derivation and details

De Sa et al. recently showed that a very general class of LDR matrices have asymptotically fast matrix-vector multiplication algorithms [14]. However, parts of the argument are left to existential results. Building upon De Sa et al. [14], we derive a simplified and self-contained algorithm for multiplication by LDR matrices with subdiagonal operators.

Since these matrices can be represented by the Krylov product formula (2), it suffices to show multiplication algorithms separately for matrix-vector multiplication by $\mathcal{K}(\mathbf{A}, \mathbf{v})^T$ and $\mathcal{K}(\mathbf{A}, \mathbf{v})$.

**Krylov transpose multiplication**  Let $\mathbf{A} \in \mathbb{R}^{n \times n}$ be a subdiagonal matrix, i.e. $\mathbf{A}_{i+1,i}$ are the only possible non-zero entries. Let $\mathbf{u}, \mathbf{v} \in \mathbb{R}^n$, we wish to compute the product $\mathcal{K}(\mathbf{A}, \mathbf{v})^T \mathbf{u}$. For simplicity assume $n$ is a power of 2.

Following [14], the vector

$$\mathbf{u}^T \mathcal{K}(\mathbf{A}, \mathbf{v}) = \begin{bmatrix} \mathbf{uv} & \mathbf{uAv} & \ldots & \mathbf{uA}^{n-1}\mathbf{v} \end{bmatrix}$$

is the coefficient vector of the polynomial in $X$

$$\mathbf{uv} + \mathbf{uAv}X + \cdots + \mathbf{uA}^{n-1}\mathbf{v}X^{n-1}$$

$$= \sum_{i=0}^{\infty} \mathbf{uA}^i X^i \mathbf{v}$$

$$= \mathbf{u}(\mathbf{I} - \mathbf{A}X)^{-1}\mathbf{v},$$

where we use the observation that $\mathbf{A}^n = 0$.

By partitioning $\mathbf{A}$ into $n/2 \times n/2$ blocks, it has the form $\begin{bmatrix} \mathbf{A}_0 & \mathbf{0} \\ a\mathbf{e}_1\mathbf{e}_{n/2}^T & \mathbf{A}_1 \end{bmatrix}$, where $\mathbf{A}_0, \mathbf{A}_1$ are subdiagonal matrices of half the size, $a$ is a scalar, and $\mathbf{e}_i$ are basis vectors. Let also $\mathbf{u}_0, \mathbf{u}_1 \in \mathbb{R}^{n/2}$, $\mathbf{v}_0, \mathbf{v}_1 \in \mathbb{R}^{n/2}$ denote the first and second halves of $\mathbf{u}, \mathbf{v}$.

By block matrix inversion for triangular matrices $\begin{bmatrix} \mathbf{A} & \mathbf{0} \\ \mathbf{C} & \mathbf{B} \end{bmatrix}^{-1} = \begin{bmatrix} \mathbf{A}^{-1} & \mathbf{0} \\ -\mathbf{B}^{-1}\mathbf{C}\mathbf{A}^{-1} & \mathbf{B}^{-1} \end{bmatrix}$, this can be written as

$$\mathbf{u}^T(\mathbf{I} - \mathbf{A}X)^{-1}\mathbf{v} = \begin{bmatrix} \mathbf{u}_0^T & \mathbf{u}_1^T \end{bmatrix} \begin{bmatrix} (\mathbf{I} - \mathbf{A}_0 X)^{-1} & \mathbf{0} \\ -(\mathbf{I} - \mathbf{A}_1 X)^{-1}(-a\mathbf{e}_1\mathbf{e}_{n/2}^T X)(\mathbf{I} - \mathbf{A}_0 X)^{-1} & (\mathbf{I} - \mathbf{A}_1 X)^{-1} \end{bmatrix} \begin{bmatrix} \mathbf{v}_0 \\ \mathbf{v}_1 \end{bmatrix}$$

$$= \mathbf{u}_0^T(\mathbf{I} - \mathbf{A}_0 X)^{-1}\mathbf{v}_0 + \mathbf{u}_1^T(\mathbf{I} - \mathbf{A}_1 X)^{-1}\mathbf{v}_1 + aX\left(\mathbf{u}_1^T(\mathbf{I} - \mathbf{A}_1 X)^{-1}\mathbf{e}_1\right)\left(\mathbf{e}_{n/2}^T(\mathbf{I} - \mathbf{A}_0 X)^{-1}\mathbf{v}_0\right)$$

Therefore $\mathbf{u}^T(\mathbf{I} - \mathbf{A}X)^{-1}\mathbf{v}$ can be computed from

$$\mathbf{u}_0^T(\mathbf{I} - \mathbf{A}_0 X)^{-1}\mathbf{v}_0 \qquad \mathbf{u}_1^T(\mathbf{I} - \mathbf{A}_1 X)^{-1}\mathbf{v}_1$$

$$\mathbf{u}_1^T(\mathbf{I} - \mathbf{A}_1 X)^{-1}\mathbf{e}_1 \qquad \mathbf{e}_{n/2}^T(\mathbf{I} - \mathbf{A}_0 X)^{-1}\mathbf{v}_0$$

with an additional polynomial multiplication and 3 polynomial addition/subtractions.

A modification of this reduction shows that the $2 \times 2$ matrix of polynomials $\begin{bmatrix} \mathbf{u} & \mathbf{e}_n \end{bmatrix}^T (\mathbf{I} - \mathbf{A}X)^{-1} \begin{bmatrix} \mathbf{v} & \mathbf{e}_1 \end{bmatrix}$ can be computed from

$$\begin{bmatrix} \mathbf{u}_0 & \mathbf{e}_n \end{bmatrix}^T (\mathbf{I} - \mathbf{A}_0 X)^{-1} \begin{bmatrix} \mathbf{v}_0 & \mathbf{e}_1 \end{bmatrix} \qquad \begin{bmatrix} \mathbf{u}_1 & \mathbf{e}_n \end{bmatrix}^T (\mathbf{I} - \mathbf{A}_1 X)^{-1} \begin{bmatrix} \mathbf{v}_1 & \mathbf{e}_1 \end{bmatrix}$$

with an additional constant number of polynomial multiplications and additions.

The complete recursive algorithm is provided in Algorithm 1, where subroutine R computes the above matrix of polynomials. For convenience, Algorithm 1 uses Python indexing notation.

A polynomial multiplication of degree $m$ in Step 8 can be computed as a convolution of size $2m$. This reduces to two Fast Fourier Transform (FFT) calls, an elementwise multiplication in the frequency domain, and an inverse FFT. The total number of calls can be further reduced to 4 FFTs and 4 inverse FFTs.

---

**Algorithm 1** Krylov Transpose (Recursive)

---

1: **function** KRYLOV_TRANSPOSE($\mathbf{A} \in \mathbb{R}^{n \times n}, \mathbf{u}, \mathbf{v} \in \mathbb{R}^n$)
2:     $\mathbf{s} \leftarrow \text{subdiagonal}(\mathbf{A})$
3:     **return** $\text{R}(\mathbf{s}, \mathbf{u}, \mathbf{v})$
4: **end function**
5: **function** $\text{R}(\mathbf{s} \in \mathbb{R}^{n-1}, \mathbf{u}, \mathbf{v})$
6:     $S_0 \leftarrow \text{R}(\mathbf{s}[0:n/2-1], \mathbf{u}[0:n/2], \mathbf{v}[0:n/2])$
7:     $S_1 \leftarrow \text{R}(\mathbf{s}[n/2:n-1], \mathbf{u}[n/2:n], \mathbf{v}[n/2:n])$
8:     $L \leftarrow \mathbf{s}[n/2-1]X \cdot \begin{bmatrix} S_1[0,1] \cdot S_0[1,0] & S_1[0,1] \cdot S_0[1,1] \\ S_1[1,1] \cdot S_0[1,0] & S_1[1,1] \cdot S_0[1,1] \end{bmatrix}$
9:     **return** $\begin{bmatrix} L[0,0] + S_0[0,0] + S_1[0,0] & L[0,1] + S_0[0,1] \\ L[1,0] + S_1[1,0] & L[1,1] \end{bmatrix}$
10: **end function**

---

Algorithm 1 defines a recursion tree, and in practice we compute this breadth first bottom-up to avoid recursive overhead. This also allows the FFT operations to be batched and computed in parallel. Thus the $d$-th layer of the algorithm (starting from the leaves) performs $\frac{n}{2^d}$ FFT computations of size $2^{d+1}$.

This completes the proof of Theorem 1.

We note several optimizations that are useful for implementation:

1. The polynomial $\mathbf{e}_n^T(\mathbf{I} - \mathbf{A}_i X)^{-1}\mathbf{e}_1$ for $i = 0, 1$ are in fact monomials, which can be shown inductively. To use the notation of Algorithm 1, $S_0[1,1]$, $S_1[1,1]$, and $L[1,1]$ are monomials. Therefore the polynomial multiplication with $S_0[1,1]$ and $S_1[1,1]$ can be done directly by coefficient-wise multiplication instead of using the FFT.

2. We don't need the polynomials $\mathbf{u}_0^T(\mathbf{I} - \mathbf{A}_0 X)^{-1}\mathbf{v}_0$ and $\mathbf{u}_1^T(\mathbf{I} - \mathbf{A}_1 X)^{-1}\mathbf{v}_1$ separately, we only need their sum. To use the notation of Algorithm 1, we don't need $S_0[0,0]$ and $S_1[0,0]$ separately, we only need their sum. In fact, by tracing the algorithm from the leaves of the recursion tree to the root, we see that across the same depth $d$, only the sum of the terms $S_0[0,0] + S_1[0,0]$ of the $n/2^d$ subproblems is required, not the individual terms. Therefore, when computing polynomial multiplication at depth $d$, we can perform the FFT of size $2^{d+1}$ and the pointwise multiplication, then sum across the $n/2^d$ problems before performing the inverse FFT of size $2^{d+1}$.

**Efficient batching with respect to input vector and rank.** Optimization 2 is especially important for efficient multiplication with respect to batched input $\mathbf{u}$ and higher rank $\mathbf{v}$. Suppose that $\mathbf{u}$ has size $n \times b$ and there are $r$ vectors $\mathbf{v}_1, \ldots, \mathbf{v}_r$, and we wish to compute $\sum_{i=1}^{r} \mathcal{K}(\mathbf{A}, \mathbf{v}_i)^T \mathbf{u}$. Naively performing Algorithm 1 on each of the $b$ inputs and each of the $r$ vectors then summing the results, takes $O(brn \log^2 n)$ time. The bottleneck of the algorithm is the polynomial multiplication $S_1[0,1] \cdot S_0[1,0]$. At depth $d$, there are $n/2^d$ subproblems, and in each of those, $S_1[0,1]$ consists of $b$ polynomials of degree at most $2^d$, while $S_0[1,0]$ consists of $r$ polynomials of degree at most $2^d$. If we apply optimization 2, we first perform the FFT of size $2^{d+1}$ on these $(b+r)n/2^d$ polynomials, then pointwise multiplication in the frequency domain to get $brn/2^d$ vectors of size $2^{d+1}$ each. Next we sum across the $n/2^d$ problems to get $br$ vectors, before performing the inverse FFT of size $2^{d+1}$ to these $br$ vectors. The summing step allows us to reduce the number of inverse FFTs from $brn/2^d$ to $br$. The total running time over all depth $d$ is then $O((b+r)n \log^2 n + brn \log n)$ instead of $O(brn \log^2 n)$.

**Krylov multiplication** De Sa et al. [14] do not provide explicit algorithms for the more complicated problem of multiplication by $\mathcal{K}(\mathbf{A}, \mathbf{v})$, instead justifying the existence of such an algorithm with the **transposition principle**. Traditional proofs of the transposition principle use circuit based arguments involving reversing arrows in the arithmetic circuit defining the algorithm's computation graph [6].

Here we show an alternative simple way to implement the transpose algorithm using any automatic differentiation (AD) implementation, which all modern deep learning frameworks include. AD

states that for any computation, its derivative can be computed with only a constant factor more operations [22].

**Proposition 6** (Transposition Principle). *If the matrix $\mathbf{M} \in \mathbb{R}^{n \times n}$ admits matrix-vector multiplication by any vector in $N$ operations, then $\mathbf{M}^T$ admits matrix-vector multiplication in $O(N + n)$ operations.*

*Proof.* Note that for any $\mathbf{x}$ and $\mathbf{y}$, the scalar $\mathbf{y}^T \mathbf{M} \mathbf{x} = \mathbf{y} \cdot (\mathbf{M}\mathbf{x})$ can be computed in $N + n$ operations.

The statement follows from applying reverse-mode AD to compute $\mathbf{M}^T \mathbf{y} = \frac{\partial}{\partial \mathbf{x}}(\mathbf{y}^T \mathbf{M} \mathbf{x})$.

Additionally, the algorithm can be optimized by choosing $\mathbf{x} = \mathbf{0}$ to construct the forward graph. $\quad\square$

To perform the optimization mentioned in Proposition 6, and avoid needing second-order derivatives when computing backprop for gradient descent, we provide an explicit implementation of non-transpose Krylov multiplication $\mathcal{K}(\mathbf{A}, \mathbf{v})$. This was found by using Proposition 6 to hand-differentiate Algorithm 1.

Finally, we comment on multiplication by the LDR-TD class. Desa et al.[14] showed that these matrices also have asymptotically efficient multiplication algorithms, of the order $O(rn \log^3 n)$ operations. However, these algorithms are even more complicated and involve operations such as inverting matrices of polynomials in a modulus. Practical algorithms for this class similar to the one we provide for LDR-SD matrices require more work to derive.

### C.3 Displacement rank and equivariance

Here we discuss in more detail the connection between LDR and equivariance. One line of work [12, 28] has used the group representation theory formalization of equivariant maps, in which the model is equivariant to a set of transformations which form a group $G$. Each transformation $g \in G$ acts on an input $x$ via a corresponding linear map $T_g$. For example, elements of the rotation group in two and three dimensions, $SO(2)$ and $SO(3)$, can be represented by 2D and 3D rotation matrices respectively. Formally, a feature map $\Phi$ is equivariant if it satisfies

$$\Phi(T_g x) = T'_g(\Phi(x)) \tag{5}$$

for representations $T, T'$ of $G$ [12, 28]. This means that perturbing the input $x$ by a transformation $g \in G$ before computing the map $\Phi$ is equivalent to first finding the features $\Phi$ and then applying the transformation. Group equivariant convolutional neural networks (G-CNNs) are a particular realization where $\Phi$ has a specific form $G \to \mathbb{R}^d$, and $T, T'$ are chosen in advance [12]. We use the notation $\mathbf{\Phi}$ to distinguish our setting, where the input $x$ is finite dimensional and $\Phi$ is linear.

**Proposition 7.** *If $\mathbf{\Phi}$ has displacement rank 0 with respect to invertible $\mathbf{A}, \mathbf{B}$, then $\mathbf{\Phi}$ is equivariant as defined by (5).*

*Proof.* Note that if $\mathbf{A}\mathbf{\Phi} = \mathbf{\Phi}\mathbf{B}$ for invertible matrices $\mathbf{A}, \mathbf{B}$ (i.e. if a matrix $\mathbf{\Phi}$ has displacement rank 0 with respect to $\mathbf{A}$ and $\mathbf{B}$), then $\mathbf{A}^i \mathbf{\Phi} = \mathbf{\Phi} \mathbf{B}^i$ also holds for $i \in \mathbb{Z}$. Also note that the set of powers of any invertible matrix forms a cyclic group, where the group operation is multiplication. The statement follows directly from this fact, where the group $G$ is $\mathbb{Z}$, and the representations $T$ and $T'$ of $G$ correspond to the cyclic groups generated by $\mathbf{A}$ and $\mathbf{B}$, respectively consisting of $\mathbf{A}^i$ and $\mathbf{B}^i$ for all $i \in \mathbb{Z}$. $\quad\square$

More generally, a feature map $\Phi$ satisfying (5) for a set of generators $S = \{g_i\}$ is equivariant with respect to the free group generated by $S$. Proposition 7 follows from the specific case of a single generator, i.e. $S = \{1\}$.

## D    Bound on VC dimension and sample complexity

In this section we upper bound the VC dimension of a neural network where all the weight matrices are LDR matrices and the activation functions are piecewise polynomials. In particular, the VC dimension is almost linear in the number of parameters, which is much smaller than the VC dimension of a network with unstructured layers. The bound on the VC dimension allows us to bound the sample

complexity to learn an LDR network that performs well among LDR networks. This formalizes the intuition that compressed parameterization reduces the complexity of the class.

**Neural network model**  Consider a neural network architecture with $W$ parameters, arranged in $L$ layers. Each layer $l$, has output dimension $n_l$, where $n_0$ is the dimension of the input data and the output dimension is $n_L = 1$. For $l = 1, \ldots, L$, let $\mathbf{i}_l \in \mathbb{R}^{n_l}$ be the input to the $l$-th layer. The input to the $(l+1)$-th layer is exactly the output of the $l$-th layer. The activation functions $\phi_l$ are piecewise polynomials with at most $p + 1$ pieces and degree at most $k \geq 1$. The input to the first layer is the data $\mathbf{i}_1 = \mathbf{x} \in \mathbb{R}^{n_1}$, and the output of the last layer is a real number $i_{L+1} \in \mathbb{R}$. The intermediate layer computation has the form:

$$i_{l+1} = \phi_l(\mathbf{M}_l \mathbf{i}_l + \mathbf{b}_l) \quad \text{(applied elementwise)}, \qquad \text{where } \mathbf{M}_l \in \mathbb{R}^{n_{l-1} \times n_l}, \ \mathbf{b}_l \in \mathbb{R}^{n_l}.$$

We assume the activation function of the final layer is the identity.

Each weight matrix $\mathbf{M}_l$ is defined through some set of parameters; for example, traditional unconstrained matrices are parametrized by their entries, and our formulation (2) is parametrized by the entries of some operator matrices $\mathbf{A}_l, \mathbf{B}_l$ and low-rank matrix $\mathbf{G}_l \mathbf{H}_l^T$. We collectively refer to all the parameters of the neural network (including the biases $b_l$) as $\theta \in \mathbb{R}^W$, where $W$ is the number of parameters.

**Bounding the polynomial degree**  The crux of the proof of the VC dimension bound is that the entries of $\mathbf{M} \in \mathbb{R}^{n \times m}$ are polynomials in terms of the entries of its parameters ($\mathbf{A}$, $\mathbf{B}$, $\mathbf{G}$, and $\mathbf{H}$). of total degree at most $c_1 m^{c_2}$ for universal constants $c_1, c_2$. This allows us to bound the total degree of all of the layers and apply Warren's lemma to bound the VC dimension.

We will first show this for the specific class of matrices that we use, where the matrix $\mathbf{M}$ is defined through equation (2).

**Lemma 1.** *Suppose that $\mathbf{M} \in \mathbb{R}^{m \times m}$ is defined as*

$$\mathbf{M} = \sum_{i=1}^{r} \mathcal{K}(\mathbf{A}, \mathbf{g}_i) \mathcal{K}(\mathbf{B}^T, \mathbf{h}_i).$$

*Then the entries of $\mathbf{M}$ are polynomials of the entries of $\mathbf{A}, \mathbf{B}, \mathbf{G}, \mathbf{H}$ with total degree at most $2m$.*

*Proof.* Since $\mathcal{K}(\mathbf{A}, \mathbf{g}_i) = \begin{bmatrix} \mathbf{g}_i & \mathbf{A}\mathbf{g}_i & \ldots & \mathbf{A}^{m-1}\mathbf{g}_i \end{bmatrix}$, and each entry of $\mathbf{A}^k$ is a polynomial of the entries of $\mathbf{A}$ with total degree at most $k$, the entries of $\mathcal{K}(\mathbf{A}, \mathbf{g}_i)$ are polynomials of the entries of $A$ and $\mathbf{g}_i$ with total degree at most $m$. Similarly the entries of $\mathcal{K}(\mathbf{B}^T, \mathbf{h}_i)$ are polynomials of the entries of $\mathbf{B}$ and $\mathbf{h}_i$ with total degree at most $m$. Hence the entries of $\mathcal{K}(\mathbf{A}, \mathbf{g}_i)\mathcal{K}(\mathbf{B}^T, \mathbf{h}_i)$ are polynomials of the entries of $\mathbf{A}, \mathbf{B}, \mathbf{G}, \mathbf{H}$ with total degree at most $2m$. We then conclude that the entries of $\mathbf{M}$ are polynomials of the entries of $\mathbf{A}, \mathbf{B}, \mathbf{G}, \mathbf{H}$ with total degree at most $2m$. $\square$

**Lemma 2.** *Suppose that the LDR weight matrices $M_l$ of a neural network have entries that are polynomials in their parameters with total degree at most $c_1 n_{l-1}^{c_2}$ for some universal constants $c_1, c_2 \geq 0$. For a fixed data point $\mathbf{x}$, at the $l$-th layer of a neural network with LDR weight matrices, each entry of $\mathbf{M}_l \mathbf{i}_l + \mathbf{b}_l$ is a piecewise polynomial of the network parameters $\theta$, with total degree at most $d_l$, where*

$$d_0 = 0, \qquad d_l = k d_{l-1} + c_1 n_{l-1}^{c_2} \quad \text{for } l = 1, \ldots, L.$$

*Thus entries of the output $\phi_l(\mathbf{M}_l \mathbf{i}_l + \mathbf{b}_l)$ are piecewise polynomials of $\theta$ with total degree at most $k d_l$. Moreover,*

$$d_l \leq c_1 k^{l-1} \sum_{j=0}^{l-1} n_j^{c_2}. \tag{6}$$

By Lemma 1, Lemma 2 applies to the specific class of matrices that we use, for $c_1 = 2$ and $c_2 = 1$. As we will see, it also applies to very general classes of structured matrices.

*Proof.* We induct on $l$. For $l = 1$, since $\mathbf{i}_1 = \mathbf{x}$ is fixed, the entries of $\mathbf{M}_1$ are polynomials of $\theta$ of degree at most $c_1 n_0^{c_2}$, and so the entries of $\mathbf{M}_1 \mathbf{i}_1 + \mathbf{b}_1$ are polynomials of $\theta$ with total degree at most $d_1 = c_1 n_0^{c_2}$. As $\phi$ is a piecewise polynomials of degree at most $k$, each entry the output

$\phi_1(\mathbf{M}_1\mathbf{i}_1 + \mathbf{b}_1)$ is a piecewise polynomial of $\theta$ with total degree at most $2n_0k$. The bound (6) holds trivially.

Suppose that the lemma is true for some $l - 1 \geq 1$. Since the entries of $\mathbf{i}_l$ are piecewise polynomials of $\theta$ with total degree at most $kd_{l-1}$ and entries of $\mathbf{M}_l$ are polynomials of $\theta$ with total degree at most $c_1 n_{l-1}^{c_2}$, the entries of $\mathbf{M}_l\mathbf{i}_l + \mathbf{b}_l$ are piecewise polynomials of $\theta$ with total degree at most $d_l = kd_{l-1} + c_1 n_{l-1}^{c_2}$. Thus $\phi_l(\mathbf{M}_l\mathbf{i}_l + \mathbf{b}_l)$ have entries that are piecewise polynomials of $\theta$ with total degree at most $kd_l$.

We can bound

$$d_l = kd_{l-1} + c_1 n_{l-1}^{c_2} \leq kc_1 k^{l-2} \sum_{j=0}^{l-2} n_j^{c_2} + c_1 n_{l-1}^{c_2} \leq c_1 k^{l-1} \sum_{j=0}^{l-1} n_j^{c_2},$$

where we have used the fact that $k \geq 1$, so $c_1 n_{l-1}^{c_2} \leq c_1 k^{l-1} n_{l-1}^{c_2}$. This concludes the proof. $\qquad\square$

**Bounding the VC dimension**  Now we are ready to bound the VC dimension of the neural network.

**Theorem 3.** *For input $x \in \mathcal{X}$ and parameter $\theta \in \mathbb{R}^W$, let $f(x, \theta)$ denote the output of the network. Let $\mathcal{F}$ be the class of functions $\{x \to f(x, \theta) \colon \theta \in \mathbb{R}^W\}$. Denote $\mathrm{sign}\,\mathcal{F} := \{x \to \mathrm{sign}\, f(x, \theta) \colon \theta \in \mathbb{R}^W\}$. Let $W_l$ be the number of parameters up to layer $l$ (i.e., the total number of parameters in layer $1, 2, \ldots, l$). Define the effective depth as*

$$\bar{L} := \frac{1}{W} \sum_{l=1}^{L} W_l,$$

*and the total number of computation units (including the input dimension) as*

$$U := \sum_{l=0}^{L} n_l.$$

*Then*
$$\mathrm{VCdim}(\mathrm{sign}\,\mathcal{F}) = O(\bar{L}W \log(pU) + \bar{L}LW \log k).$$
*In particular, if $k = 1$ (corresponding to piecewise linear networks) then*
$$\mathrm{VCdim}(\mathrm{sign}\,\mathcal{F}) = O(\bar{L}W \log(pU)) = O(LW \log W).$$

We adapt the proof of the upper bound from Bartlett et al. [4], Harvey et al. [24]. The main technical tool is Warren's lemma [48], which bounds the growth function of a set of polynomials. We state a slightly improved form here from Anthony and Bartlett [3, Theorem 8.3].

**Lemma 3.** *Let $p_1, \ldots, p_m$ be polynomials of degree at most $d$ in $n \leq m$ variables. Define*
$$K := |\{(\mathrm{sign}(p_1(\mathbf{x})), \ldots, \mathrm{sign}(p_m(\mathbf{x}))) : \mathbf{x} \in \mathbb{R}^n\}|,$$
*i.e., $K$ is the number of possible sign vectors given by the polynomials. Then $K \leq 2(2emd/n)^n$.*

*Proof of Theorem 3.*  Fixed some large integer $m$ and some inputs $\mathbf{x}_1, \ldots, \mathbf{x}_m$. We want to bound the number of sign patterns that the neural network can output for the set of input $\mathbf{x}_1, \ldots, \mathbf{x}_m$:
$$K := \left| \{(\mathrm{sign}\, f(\mathbf{x}_1, \theta), \ldots, \mathrm{sign}\, f(\mathbf{x}_m, \theta)) \colon \theta \in \mathbb{R}^W\} \right|.$$

We want to partition the parameter space $\mathbb{R}^W$ so that for a fixed $\mathbf{x}_j$, the output $f(\mathbf{x}_j, \theta)$ is a polynomial on each region in the partition. Then we can apply Warren's lemma to bound the number of sign patterns. Indeed, for any partition $\mathcal{S} = \{P_1, \ldots, P_N\}$ of the parameter space $\mathbb{R}^W$, we have

$$K \leq \sum_{j=1}^{N} |\{(\mathrm{sign}\, f(\mathbf{x}_1, \theta), \ldots, \mathrm{sign}\, f(\mathbf{x}_m, \theta)) \colon \theta \in P_j\}|. \tag{7}$$

We construct the partitions iteratively layer by layer, through a sequence $\mathcal{S}_0, \mathcal{S}_1, \ldots, \mathcal{S}_{L-1}$ of successive refinements, satisfying two properties:

1. $|\mathcal{S}_0| = 1$ and for each $1 \leq l \leq L - 1$,

$$|\mathcal{S}_l| \leq |\mathcal{S}_{l-1}| \, 2 \left( \frac{2empn_l d_l}{W_l} \right)^{W_l},$$

where $n_l$ is the dimension of the output of the $l$-th layer, $d_l$ is the bound on the total degree of $\mathbf{M}_l \mathbf{i}_l + \mathbf{b}_l$ as piecewise polynomials of $\theta$ as defined in Lemma 2, and $W_l$ is the number of parameters up to layer $l$ (i.e., the total number of parameters in layer $1, 2, \ldots, l$).

2. For each $l = 0, \ldots, L - 1$, for each element $S$ of $\mathcal{S}_l$, for each fixed data point $\mathbf{x}_j$ (with $j = 1, \ldots, m$), the entries of the output $\phi_l(\mathbf{M}_l \mathbf{i}_l + \mathbf{b}_l)$ when restricted to $S$ are polynomials of $\theta$ with total degree at most $kd_{l-1}$.

We can define $\mathcal{S}_0 = \mathbb{R}^W$, which satisfies property 2, since at layer 1, the entries of $\mathbf{i}_1 = \mathbf{x}_j$ (for fixed $\mathbf{x}_j$) are polynomials of $\theta$ of degree $d_0 = 0$.

Suppose that we have constructed $\mathcal{S}_0, \ldots, \mathcal{S}_{l-1}$, and we want to define $\mathcal{S}_l$. For any $h \in [n_l], j \in [m]$, and $S \in \mathcal{S}_{l-1}$, let $p_{h,\mathbf{x}_j,S}(\theta) = (\mathbf{M}_l \mathbf{i}_l + \mathbf{b}_l)_h|_S$ be the $h$-th entry of $\mathbf{M}_l \mathbf{i}_l + \mathbf{b}_l$ restricted to the region $S$. By the inductive hypothesis, for each $S \in \mathcal{S}_{l-1}$, the entries of $i_l$ when restricted to $S$ are polynomials of $\theta$ of total degree at most $kd_{l-1}$. Thus by Lemma 2, the entries of $\mathbf{M}_l \mathbf{i}_l + \mathbf{b}_l$ when restricted to $S$ are polynomials of $\theta$ with total degree at most $kd_{l-1} + c_1 n_{l-1}^{c_2} = d_l$, and depends on at most $W_l$ many variables.

Since the activation function is piecewise polynomial with at most $p$ pieces, let $\{t_1, \ldots, t_p\}$ be the set of breakpoints. For any fixed $S \in \mathcal{S}_{l-1}$, by Lemma 3, the polynomials

$$\left\{ p_{h,\mathbf{x}_j,S}(\theta) - t_i : h \in [n_l], j \in [m], i \in [p] \right\}$$

can have at most

$$\Pi := 2 \left( \frac{2e(n_l m p) d_l}{W_l} \right)^{W_l}$$

distinct sign patterns when $\theta \in \mathbb{R}^W$. We can then partition $\mathbb{R}^W$ into this many regions so that within each region, all these polynomials have the same signs. Intersecting all these regions with $S$ yields a partition of $S$ into at most $\Pi$ subregions. Applying this for all $S \in \mathcal{S}_{l-1}$ gives a partition $\mathcal{S}_l$ that satisfies the property 1.

Fix some $S' \in \mathcal{S}_n$. When $\theta$ is restricted to $S'$, by construction, all the polynomials

$$\left\{ p_{h,\mathbf{x}_j,S}(\theta) - t_i : h \in [n_l], j \in [m], i \in [p] \right\}$$

have the same sign. This means that the entries of $\mathbf{M}_l \mathbf{i}_l + \mathbf{b}_l$ lie between two breakpoints of the activation function, and so the entries of the output $\phi_l(\mathbf{M}_l \mathbf{i}_l + \mathbf{b}_l)$ are fixed polynomials in $W_l$ variables of degree at most $kd_l$.

By this recursive construction, $\mathcal{S}_{L-1}$ is a partition of $\mathbb{R}^W$ such that for $S \in \mathcal{S}_{L-1}$ the network output for any input $\mathbf{x}_j$ is a fixed polynomial of $\theta \in S$ of degree at most $kd_{L-1} + c_1 n_{L-1}^{c_2} = d_L$ (recall that we assume the activation function of the final layer is the identity). Hence we can apply Lemma 3 again:

$$|\{(\text{sign } f(\mathbf{x}_1, \theta), \ldots, \text{sign } f(\mathbf{x}_m, \theta)) : \theta \in S\}| \leq 2 \left( \frac{2emkd_L}{W_L} \right)^{W_L}.$$

By property 1, we can bound the size of $\mathcal{S}_{L-1}$:

$$|\mathcal{S}_L| \leq \prod_{l=1}^{L-1} 2 \left( \frac{2emn_l p d_l}{W_l} \right)^{W_l}.$$

Combining the two bounds along with equation (7) yields

$$K \leq \prod_{l=1}^{L} 2 \left( \frac{2empn_l d_l}{W_l} \right)^{W_l}.$$

We can take logarithm and apply Jensen's inequality, with $\bar{W} := \sum_{l=1}^{L} W_l$:

$$\log_2 K \leq L + \sum_{l=1}^{L} W_l \log_2 \frac{2empn_l d_l}{W_l}$$

$$= L + \bar{W} \sum_{l=1}^{L} \frac{W_l}{\bar{W}} \log_2 \frac{2empn_l d_l}{W_l}$$

$$\leq L + \bar{W} \log_2 \left( \sum_{l=1}^{L} \frac{W_l}{\bar{W}} \frac{2empn_l d_l}{W_l} \right) \qquad \text{(Jensen's inequality)}$$

$$= L + \bar{W} \log_2 \frac{2emp \sum_{l=1}^{L} n_l d_l}{\bar{W}}.$$

We can bound $\sum n_l d_l$ using the bound on $d_l$ from Lemma 2:

$$\sum_{l=1}^{L} n_l d_l \leq \sum_{l=1}^{L} n_l c_1 k^{l-1} \sum_{j=0}^{l-1} n_j^{c_2} \leq LU c_1 k^{L-1} U^{c_2} \leq c_1 U^{c_2+2} k^L,$$

where we used the fact that $L \leq U$. Thus

$$\log_2 K \leq L + \bar{W} \log_2 \frac{2c_1 emp U^{2+c_2} k^L}{\bar{W}}.$$

To bound the VC-dimension, recall that by definition, if $\text{VCdim}(\text{sign}\,\mathcal{F}) = m$ then exists $m$ data points $\mathbf{x}_1, \ldots, \mathbf{x}_m$ such that the output of the model can have $2^n$ sign patterns. The bound on $\log_2 K$ then implies

$$\text{VCdim}(\text{sign}\,\mathcal{F}) \leq L + \bar{W} \log_2 \frac{2c_1 ep U^{2+c_2} k^L \text{VCdim}(\text{sign}\,\mathcal{F})}{\bar{W}}.$$

We then use Lemma 4 below, noting that $2c_1 ep U^{2+c_2} k^L \geq 16$, to conclude that

$$\text{VCdim}(\text{sign}\,\mathcal{F}) \leq L + \bar{W} \log_2(2c_1 ep U^{2+c_2} k^L \log_2(2c_1 ep U^{2+c_2} k^L)) = O(\bar{L}W \log(pU) + \bar{L}LW \log k),$$

completing the proof.

$\square$

A bound on the VC dimension immediate yields a bound on the sample complexity of learning from this class of neural networks with LDR matrices [47].

**Corollary 2.** *The class of neural network with LDR matrices as weights and piecewise linear activation is $(\epsilon, \delta)$-PAC-learnable with a sample of size*

$$O \left( \frac{LW \log W + \log \frac{1}{\delta}}{\epsilon} \right).$$

Since the number of parameters $W$ is around the square root of the number of parameters of a network with unstructured layers (assuming fixed rank of the LDR matrices), the sample complexity of LDR networks is much smaller than that of general unstructured networks.

**Lemma 4** (Lemma 16 of [24]). *Suppose that $2^m \leq 2^t(mr/w)^w$ for some $r \geq 16$ and $m \geq w \geq t \geq 0$. Then, $m \leq t + w \log_2(2r \log_2 r)$.*

**Extension to rational functions.** We now show that Theorem 3 holds for matrices where the entries are rational functions—rather than polynomials—of its parameters, incurring only a constant in the bound. To define the function class $\text{sign}\,\mathcal{F}$, we account for the possibility of poles by defining $\text{sign}(a/0) = 0$.

We only need to check that Lemma 2 and Lemma 3 still hold when polynomials are replaced by rational functions everywhere, and the degree of a rational function is defined as the usual $\deg(a/b) = \max\{\deg a, \deg b\}$. To show Lemma 2 still holds, it suffices that the compositional degree bound

$\deg(f \circ g) \le \deg(f)\deg(g)$ holds for rational functions $f, g$, just as in the polynomial case. To show Lemma 3 in the case when $p_i = a_i/b_i$ are rational functions, we note that $\text{sign}(p_i(x)) = \text{sign}(a_i(x)b_i(x))$, and furthermore $\deg(a_ib_i) \le 2\deg(p_i)$. Appealing to the polynomial version of Lemma 3 shows that it holds in the rational function setting with a slightly weaker upper bound $K \le 2(4emd/n)^n$. This gets converted to a constant factor in the result of Theorem 3.

Next, we extend Lemma 1 by showing that generic LDR matrices have entries which are rational functions of their parameters. This immediately lets us conclude that neural networks built from any LDR matrices satisfy the VC dimension bounds of Theorem 3.

**Lemma 5.** *If* $\mathbf{M} \in \mathbb{R}^{m \times m}$ *satisfies* $\mathbf{AM} - \mathbf{MB} = \mathbf{GH}^T$, *then the entries of* $\mathbf{M}$ *are rational functions of the entries of* $\mathbf{A}, \mathbf{B}, \mathbf{G}, \mathbf{H}$ *with total degree at most* $c_1 m^{c_2}$ *for some universal constants* $c_1, c_2 > 0$.

*Proof.* The vectorization of the Sylvester equation $\mathbf{AM} - \mathbf{MB} = \mathbf{R}$ is $(\mathbf{I} \otimes \mathbf{A} - \mathbf{B}^\top \otimes \mathbf{I})\text{vec}(\mathbf{M}) = \text{vec}(\mathbf{R})$, where $\text{vec}$ denotes the vectorization operation by stacking a matrix's columns, and $\otimes$ is the Kronecker product. Note that the entries of $\mathbf{N}^{-1}$ for an arbitrary matrix $\mathbf{N} \in \mathbb{R}^{n \times n}$ are rational functions with degree $n$ in the entries of $\mathbf{N}$, and $\mathbf{R} = \mathbf{GH}^\top$ has degree 2 in the entries of $\mathbf{G}, \mathbf{H}$. Therefore the entries of

$$\text{vec}(\mathbf{M}) = (\mathbf{I} \otimes \mathbf{A} - \mathbf{B}^\top \otimes \mathbf{I})^{-1}\text{vec}(\mathbf{R})$$

have degree $n^2 + 2$ in the entries of $\mathbf{A}, \mathbf{B}, \mathbf{G}, \mathbf{H}$. $\square$

Note that many other classes of matrices satisfy this lemma. For example, a large class of matrices satisfying a property called *low recurrence width* was recently introduced as a way of generalizing many known structured matrices [14]. The low recurrence width matrices are explicitly defined through a polynomial recurrence and satisfy the bounded degree condition. Additionally, Lemma 5 holds when the parameters $\mathbf{A}, \mathbf{B}$ themselves are structured matrices with entries having polynomial degree in terms of some parameters. This includes the case when they are quasiseparable matrices, the most general class of LDR previously analyzed [14].

# E  Additional results

## E.1  Additional baselines and comparisons at multiple budgets

In Tables 6 and 7 we compare to baselines at parameter budgets corresponding to both the LDR-TD and LDR-SD classes in the SHL and CNN models. In Tables 8 and 9, we also compare to two additional baselines, network pruning [23] and a baseline used in [7], in which the number of hidden units is reduced to meet the parameter budget. We refer to this baseline as RHU ("reduced hidden units"). We show consistent improvements of LDR-SD over both methods at several budgets. We note that unlike the structured matrix methods which provide compression benefits during both training and inference, pruning requires first training the original model, followed by retraining with a fixed sparsity pattern.

## E.2  Sample complexity and generalization

As shown in Tables 10 and 11, we investigated how the performance of the structured and general unstructured fully-connected layers varied with the amount of training data. On the MNIST variants, we trained both the single hidden layer and CNN models with random subsamples of 25%, 50%, and 75% of the training set, with 15% of the training set used for validation in all settings. In addition, in Table 12, we compare the generalization error of structured classes with an unstructured model, and find that the structured classes have consistently lower generalization error.

## E.3  Additional visualizations

In Figure 6, we visualize the learned subdiagonal on NORB along with images from the dataset.

On the MNIST-bg-rot dataset [30], we note that Chen et al. [7] also tested several methods on this dataset, including Random Edge Removal [11], Low Rank Decomposition [15], Dark Knowledge [25], HashedNets [7], and HashedNets with Dark Knowledge, and reported test errors of 73.17, 80.63, 79.03, 77.40, 59.20, and 58.25, where each method had 12406 parameters in the architecture. We

Table 6: Test accuracy when replacing the hidden layer with structured classes in the **single hidden layer** architecture, at parameter budgets corresponding to LDR-TD and LDR-SD rank one. Rank is in parentheses. The first group of structured methods (in orange) all have compression factors (relative to a general unstructured layer) of 98 on MNIST-bg-rot and MNIST-noise, and 128 on CIFAR-10 and NORB. The second group of structured methods (in blue) all have compression factors of 196 on MNIST-bg-rot and MNIST-noise, and 256 on CIFAR-10 and NORB.

| Method | MNIST-bg-rot | MNIST-noise | CIFAR-10 | NORB |
|---|---|---|---|---|
| Unstructured | 44.08 | 65.15 | 46.03 | 59.83 |
| LDR-TD ($r = 1$) | **45.81** | **78.45** | **45.33** | **62.75** |
| Toeplitz-like [45] ($r = 4$) | 42.67 | 75.75 | 41.78 | 59.38 |
| Hankel-like ($r = 4$) | 42.23 | 73.65 | 41.40 | 60.09 |
| Vandermonde-like ($r = 4$) | 37.14 | 59.80 | 33.93 | 48.98 |
| Low-rank [15] ($r = 4$) | 35.67 | 52.25 | 32.28 | 43.66 |
| LDR-SD ($r = 1$) | **44.74** | **78.80** | **43.29** | **63.78** |
| Toeplitz-like [45] ($r = 2$) | 42.07 | 74.25 | 40.68 | 57.27 |
| Hankel-like ($r = 2$) | 41.01 | 71.20 | 40.46 | 57.95 |
| Vandermonde-like ($r = 2$) | 33.56 | 50.85 | 28.99 | 43.21 |
| Low-rank [15] ($r = 2$) | 32.64 | 38.85 | 24.93 | 37.03 |

Table 7: Test accuracy when replacing the fully-connected layer with structured classes in the **CNN** architecture, at parameter budgets corresponding to LDR-TD and LDR-SD rank one. Rank is in parentheses. The first group of structured methods (in orange) all have compression factors (relative to a general unstructured layer) of 98 on MNIST-bg-rot and MNIST-noise, and 128 on CIFAR-10 and NORB. The second group of structured methods (in blue) all have compression factors of 196 on MNIST-bg-rot and MNIST-noise, and 256 on CIFAR-10 and NORB.

| Method | MNIST-bg-rot | MNIST-noise | CIFAR-10 | NORB |
|---|---|---|---|---|
| Fully-connected | 67.94 | 90.30 | 68.09 | 75.16 |
| LDR-TD ($r = 1$) | **68.79** | **92.55** | 66.63 | **74.23** |
| Toeplitz-like [45] ($r = 4$) | 63.23 | 91.60 | 67.10 | 72.25 |
| Hankel-like ($r = 4$) | 64.21 | 90.80 | **68.10** | 71.23 |
| Vandermonde-like ($r = 4$) | 61.76 | 90.40 | 63.63 | 72.11 |
| Low-rank [15] ($r = 4$) | 60.35 | 87.30 | 60.90 | 71.47 |
| LDR-SD ($r = 1$) | **67.40** | **92.20** | 65.48 | **73.63** |
| Toeplitz-like [45] ($r = 2$) | 63.63 | 91.45 | 67.15 | 71.64 |
| Hankel-like ($r = 2$) | 64.08 | 90.65 | **67.49** | 71.21 |
| Vandermonde-like ($r = 2$) | 51.38 | 86.50 | 58.00 | 68.08 |
| Low-rank [15] ($r = 2$) | 41.91 | 71.15 | 48.48 | 65.34 |

(a) Subdiagonal of **B** (NORB)

(b) Images from NORB

Figure 6: We visualize the learned subdiagonal of the operator **B** and images from the NORB dataset. We observe a centering phenomenon similar to that described in Figure 4.

found that our LDR-SD class, with 10986 parameters in the architecture, achieved a test error of 55.26, as shown in Table 6, outperforming all methods evaluated by Chen et al. [7]. Sindhwani et al. [45]

Table 8: On the MNIST variants, in the **single hidden layer** architecture, we compare LDR-SD, pruning [23], and a baseline which reduces the number of hidden units (denoted RHU), at multiple budgets. At each budget, we adjust the number of pruned weights or hidden units to match as closely as possible the parameter budget of LDR-SD. Parameter counts of fully-connected layers for LDR-SD and pruning at ranks 1,2,4,8,12, and 16 are 10986, 12554, 15690, 21962, 28234, and 34506 respectively, and 11126, 12714, 15890, 22242, 28594, 34946 for RHU (for which parameter count cannot be controlled exactly). As shown above, we find that the classification accuracy of LDR-SD consistently exceeds that of both methods.

| Rank of LDR-SD | LDR-SD | Pruning [23] | RHU [7] |
|---|---|---|---|
| 1 | **44.74** | 40.41 | 37.18 |
| 2 | **44.46** | 41.18 | 37.60 |
| 4 | **47.72** | 42.45 | 37.98 |
| 8 | **48.76** | 43.52 | 39.77 |
| 12 | **48.90** | 43.19 | 40.56 |
| 16 | **49.51** | 43.58 | 40.70 |

(a) MNIST-bg-rot

| Rank of LDR-SD | LDR-SD | Pruning [23] | RHU [7] |
|---|---|---|---|
| 1 | **78.80** | 67.75 | 62.85 |
| 2 | **77.95** | 69.35 | 62.55 |
| 4 | **78.32** | 68.25 | 63.40 |
| 8 | **78.63** | 67.25 | 64.45 |
| 12 | **78.33** | 67.30 | 63.85 |
| 16 | **78.08** | 66.95 | 66.10 |

(b) MNIST-noise

Table 9: On the MNIST variants, in the **CNN** architecture, we compare LDR-SD, pruning [23], and a baseline which reduces the number of hidden units (denoted RHU), at multiple budgets. At each budget, we adjust the number of pruned weights or hidden units to match as closely as possible the parameter budget of LDR-SD. Parameter counts of fully-connected layers for LDR-SD and pruning at ranks 1,2,4,8,12, and 16 are 11770, 13338, 16474, 22746, 29018, and 35290 respectively, and 11935, 13525, 16705, 23065, 29425, 35785 for RHU (for which parameter count cannot be controlled exactly). As shown above, we find that the classification accuracy of LDR-SD consistently exceeds that of both methods.

| Rank of LDR-SD | LDR-SD | Pruning [23] | RHU [7] |
|---|---|---|---|
| 1 | **67.40** | 64.25 | 64.03 |
| 2 | **67.53** | 64.05 | 64.67 |
| 4 | **67.96** | 65.50 | 66.37 |
| 8 | **67.21** | 64.12 | 64.70 |
| 12 | **68.54** | 65.65 | 65.99 |
| 16 | **67.00** | 65.59 | 66.47 |

(a) MNIST-bg-rot

| Rank of LDR-SD | LDR-SD | Pruning [23] | RHU [7] |
|---|---|---|---|
| 1 | **92.20** | 90.80 | 90.95 |
| 2 | **92.75** | 91.65 | 91.00 |
| 4 | **91.30** | 90.60 | 91.25 |
| 8 | **91.95** | 91.05 | 90.65 |
| 12 | **92.10** | 90.00 | 90.85 |
| 16 | **93.20** | 90.55 | 90.40 |

(b) MNIST-noise

Table 10: On the MNIST variants, in the **single hidden layer** architecture, we show how the number of training samples affects the performance of the unstructured model and the structured classes. Columns correspond to models trained on 25%, 50%, 75% and 100% of the training data (randomly subsampled). LDR-TD and LDR-SD consistently outperform the structured baselines at the tested subsampling ratios. On MNIST-bg-rot, LDR-TD only needs 75% of the training data to outperform the unstructured model trained on 100% of the training data. On MNIST-noise, both LDR-TD and LDR-SD only need 25% of the training data to outperform the unstructured layer. All are rank one.

| Method | 25% | 50% | 75% | 100% | Method | 25% | 50% | 75% | 100% |
|---|---|---|---|---|---|---|---|---|---|
| Unstructured | 34.46 | 38.80 | 43.35 | 44.08 | Unstructured | 59.30 | 61.85 | 65.35 | 65.15 |
| LDR-TD | 34.01 | 39.59 | **44.35** | **45.81** | LDR-TD | 65.45 | **74.60** | **77.45** | 78.45 |
| LDR-SD | **35.64** | **39.78** | 42.72 | 44.74 | LDR-SD | 67.90 | 71.15 | 76.95 | **78.80** |
| Toeplitz-like | 33.71 | 36.44 | 39.32 | 41.12 | Toeplitz-like | 56.15 | 67.75 | 72.30 | 73.95 |
| Low-rank | 21.44 | 23.46 | 23.48 | 25.06 | Low-rank | 24.25 | 26.20 | 26.85 | 26.40 |

|         (a) MNIST-bg-rot          |         (b) MNIST-noise          |

Table 11: On the MNIST variants, in the **CNN** architecture, we show how the number of training samples affects the performance of the unstructured model and the structured classes. Columns correspond to models trained on 25%, 50%, 75% and 100% of the training data (randomly subsampled). LDR-TD and LDR-SD consistently outperform the structured baselines at the tested subsampling ratios. On MNIST-noise, both LDR-TD and LDR-SD only need 50% of the training data to outperform the unstructured layer. All are rank one.

| Method | 25% | 50% | 75% | 100% | Method | 25% | 50% | 75% | 100% |
|---|---|---|---|---|---|---|---|---|---|
| Unstructured | 54.12 | 62.53 | 67.52 | 67.94 | Unstructured | 81.85 | 88.25 | 89.75 | 90.30 |
| LDR-TD | **53.66** | **62.15** | **67.25** | **68.79** | LDR-TD | 86.45 | **91.35** | **93.00** | **92.55** |
| LDR-SD | 50.72 | 61.92 | 65.93 | 67.40 | LDR-SD | **86.95** | 90.90 | 91.55 | 92.20 |
| Toeplitz-like | 49.10 | 57.20 | 61.53 | 63.00 | Toeplitz-like | 81.65 | 88.15 | 90.90 | 90.95 |
| Low-rank | 26.98 | 27.97 | 28.97 | 29.63 | Low-rank | 33.15 | 38.40 | 42.55 | 44.55 |

|         (a) MNIST-bg-rot          |         (b) MNIST-noise          |

Table 12: Generalization error for unstructured, LDR-TD, LDR-SD, Toeplitz-like, low-rank classes on the single hidden layer architecture. Consistent with Theorem 2, the structured classes have consistently lower generalization error than the unstructured model. All are rank one.

| Method | MNIST-bg-rot | MNIST-noise | CIFAR-10 | NORB |
|---|---|---|---|---|
| Unstructured | 55.78 | 21.63 | 34.32 | 40.03 |
| LDR-TD | 13.52 | 11.36 | 7.10 | 9.51 |
| LDR-SD | 12.87 | 12.65 | 6.29 | 8.68 |
| Toeplitz-like [45] | 7.98 | 15.80 | 5.59 | 7.87 |
| Low-rank [15] | 8.40 | 0.31 | 0.09 | 2.59 |

later also tested on this dataset, and reported test errors of 68.4, 62.11, and 55.21 for Fastfood (10202 parameters), Circulant (8634 parameters), and Toeplitz-like, $r = 2$ (10986 parameters). LDR-SD exceeds their reported results for Fastfood and Circulant [8], but not that of Toeplitz-like. We did find that our proposed classes consistently exceeded the performance of our own implementation of Toeplitz-like on this dataset (Table 1, Figure 3, and Tables 6 and 7).

## E.4 Rectangles dataset

We provide an interesting example of a case where LDR-TD and LDR-SD do not exceed the performance of the fixed operator classes in the single hidden layer architecture. In this simple dataset from Larochelle et al. [30], the task is to classify a binary image of a rectangle as having a greater length or width. We show examples of the dataset in Figure 7. On this dataset, in contrast to the more

challenging datasets (MNIST-bg-rot [30], MNIST-noise [30], CIFAR-10 [29], and NORB [32]) we tested on, every structured class outperforms an unconstrained model (622506 parameters), including the circulant class [8] which compresses the hidden layer by 784x, and expanding the class beyond Toeplitz-like does not improve performance. We hypothesize that this is because the Toeplitz-like class may enforce the right structure, in the sense that it is sufficiently expressive to fit a perfect model on this dataset, but not expansive enough to lead to overfitting. For example, while the Toeplitz-like operators model approximate shift equivariance (discussed in Section 4 and Proposition 7 in Section C.3), the additional scaling that subdiagonal operators provide is unnecessary on these binary inputs.

Figure 7: Examples of images from the rectangles dataset [30].

Table 13: Test accuracy when replacing the hidden layer with structured classes on the rectangles dataset [30]. Where applicable, rank ($r$) is in parentheses, and the number of parameters in the architecture is in italics below each method.

| Method | Test Accuracy |
|---|---|
| Unconstrained | 91.94 |
| | *622506* |
| LDR-TD ($r = 1$) | 98.53 |
| | *14122* |
| LDR-SD ($r = 1$) | 98.39 |
| | *10986* |
| Toeplitz-like ($r = 4$) [45] | **99.29** |
| | *14122* |
| Hankel-like ($r = 4$) | 97.77 |
| | *14122* |
| Vandermonde-like ($r = 4$) | 94.11 |
| | *14122* |
| Low-rank ($r = 4$) [15] | 92.80 |
| | *14122* |
| Fastfood [49] | 92.20 |
| | *10202* |
| Circulant [8] | 95.58 |
| | *8634* |

### E.5 Acceleration at inference time

We empirically study the acceleration obtained at inference time (on CPU) with our implementation of the algorithms for multiplication by LDR-SD described in Appendix C.2. We generated random parameters for each class and ran each multiplication algorithm 1000 times to compare the speedup of each class over an unstructured multiply. Each test was repeated 10 times, and the minimum total runtime over the 10 tests was used for each class. As shown in Figure 8 and Table 14, at $n \geq 4096$, our simple Python implementation is 3.34-46.06x faster than the highly optimized unstructured matrix-vector multiply (a BLAS level 2 operation). We also compare with two other structured classes, low-rank and Toeplitz-like, at $r = 1, 2, 4, 8, 16$. A batch size of one was used in all tests. The time complexity of multiplication by low-rank and Toeplitz-like is $O(nr)$ and $O(nr \log n)$ respectively, compared to $O(nr \log^2 n)$ for LDR-SD.

Table 14: Acceleration of $n \times n$ structured classes over unstructured matrix-vector multiply at inference time. Experimental details are in Appendix E.5.

| | Rank | | | | |
| --- | --- | --- | --- | --- | --- |
| $n$ | 1 | 2 | 4 | 8 | 16 |
| $2^9$ | $5.15 \times 10^1$ | $2.43 \times 10^1$ | $2.46 \times 10^1$ | $2.08 \times 10^1$ | $1.81 \times 10^1$ |
| $2^{10}$ | $1.39 \times 10^2$ | $5.41 \times 10^1$ | $5.66 \times 10^1$ | $4.62 \times 10^1$ | $3.43 \times 10^1$ |
| $2^{11}$ | $4.14 \times 10^2$ | $1.60 \times 10^2$ | $1.71 \times 10^2$ | $1.05 \times 10^2$ | $6.90 \times 10^1$ |
| $2^{12}$ | $2.38 \times 10^3$ | $8.71 \times 10^2$ | $7.46 \times 10^2$ | $4.73 \times 10^2$ | $3.59 \times 10^2$ |
| $2^{13}$ | $5.96 \times 10^3$ | $1.75 \times 10^3$ | $1.65 \times 10^3$ | $1.13 \times 10^3$ | $8.86 \times 10^2$ |
| $2^{14}$ | $8.35 \times 10^3$ | $3.44 \times 10^3$ | $3.40 \times 10^3$ | $2.29 \times 10^3$ | $1.74 \times 10^3$ |
| $2^{15}$ | $1.79 \times 10^4$ | $7.50 \times 10^3$ | $7.53 \times 10^3$ | $4.91 \times 10^3$ | $3.70 \times 10^3$ |

(a) Low-rank

| | Rank | | | | |
| --- | --- | --- | --- | --- | --- |
| $n$ | 1 | 2 | 4 | 8 | 16 |
| $2^9$ | $3.06 \times 10^{-1}$ | $2.60 \times 10^{-1}$ | $2.32 \times 10^{-1}$ | $1.86 \times 10^{-1}$ | $1.61 \times 10^{-1}$ |
| $2^{10}$ | $7.34 \times 10^{-1}$ | $6.21 \times 10^{-1}$ | $5.18 \times 10^{-1}$ | $4.00 \times 10^{-1}$ | $3.28 \times 10^{-1}$ |
| $2^{11}$ | $1.90 \times 10^0$ | $1.71 \times 10^0$ | $1.38 \times 10^0$ | $1.08 \times 10^0$ | $8.46 \times 10^{-1}$ |
| $2^{12}$ | $1.23 \times 10^1$ | $1.01 \times 10^1$ | $7.92 \times 10^0$ | $5.97 \times 10^0$ | $4.62 \times 10^0$ |
| $2^{13}$ | $3.34 \times 10^1$ | $2.73 \times 10^1$ | $2.26 \times 10^1$ | $1.52 \times 10^1$ | $1.23 \times 10^1$ |
| $2^{14}$ | $6.96 \times 10^1$ | $5.68 \times 10^1$ | $4.19 \times 10^1$ | $3.00 \times 10^1$ | $2.26 \times 10^1$ |
| $2^{15}$ | $1.49 \times 10^2$ | $1.19 \times 10^2$ | $9.07 \times 10^1$ | $5.46 \times 10^1$ | $3.82 \times 10^1$ |

(b) Toeplitz-like

| | Rank | | | | |
| --- | --- | --- | --- | --- | --- |
| $n$ | 1 | 2 | 4 | 8 | 16 |
| $2^9$ | $6.68 \times 10^{-2}$ | $4.63 \times 10^{-2}$ | $4.05 \times 10^{-2}$ | $3.10 \times 10^{-2}$ | $2.56 \times 10^{-2}$ |
| $2^{10}$ | $1.49 \times 10^{-1}$ | $1.20 \times 10^{-1}$ | $9.45 \times 10^{-2}$ | $6.73 \times 10^{-2}$ | $5.24 \times 10^{-2}$ |
| $2^{11}$ | $4.99 \times 10^{-1}$ | $4.32 \times 10^{-1}$ | $3.02 \times 10^{-1}$ | $1.94 \times 10^{-1}$ | $1.37 \times 10^{-1}$ |
| $2^{12}$ | $3.34 \times 10^0$ | $2.57 \times 10^0$ | $1.61 \times 10^0$ | $1.06 \times 10^0$ | $7.52 \times 10^{-1}$ |
| $2^{13}$ | $9.71 \times 10^0$ | $6.61 \times 10^0$ | $4.40 \times 10^0$ | $2.46 \times 10^0$ | $1.68 \times 10^0$ |
| $2^{14}$ | $2.12 \times 10^1$ | $1.41 \times 10^1$ | $8.38 \times 10^0$ | $4.35 \times 10^0$ | $3.00 \times 10^0$ |
| $2^{15}$ | $4.61 \times 10^1$ | $2.82 \times 10^1$ | $1.60 \times 10^1$ | $8.58 \times 10^0$ | $5.70 \times 10^0$ |

(c) LDR-SD

# F  Experimental details

## F.1  Image classification

In Table 15, we provide details on the datasets we use for evaluation. For all our experiments, batch sizes were chosen to be 50. NORB was downsampled to $32 \times 32$, and the left stereo image was used. Training was performed with stochastic gradient descent with momentum, with the number of epochs set to 50 on all datasets. 15% of the training data was used for the validation set on all experiments. We fixed momentum at 0.9 for all methods for all experiments, and performed a grid search over learning rate. Unless otherwise stated, for each method, we tested the learning rates {0.0002, 0.0005, 0.001, 0.002}, with three trials (with random initializations) per learning rate. For each trial, we test on the validation set at each epoch, and report the test accuracy of the model with the highest validation accuracy, over all learning rates, trials, and epochs.

Figure 8: Acceleration of $n \times n$ structured classes over unstructured matrix-vector multiply at inference time. At $n \geq 4096$, LDR-SD ($r = 1$) achieves a speedup of 3.34-46.06x over unstructured. Data for higher ranks are shown in Table 14. The comparison to the low-rank and Toeplitz-like classes illustrates a tradeoff involved in broadening the class of structured matrices we learn over. Though LDR-SD consistently outperforms these classes on downstream quality, its computational cost of multiplication is $O(nr \log^2 n)$, compared to $O(nr)$ and $O(nr \log n)$ for low-rank and Toeplitz-like respectively. Experimental details are in Appendix E.5.

In Figure 3, for each method and each of the four learning rates, we perform five trials with random initializations and report the average and standard deviation of the test accuracy of the learning rate with the highest average validation accuracy.

Table 15: Overview of the image classification datasets used in this work. For all datasets, 15% of the training set was used for the validation set.

| Dataset | Training Examples | Test Examples | Number of Classes |
|---|---|---|---|
| MNIST-bg-rot [30] | 12000 | 50000 | 10 |
| MNIST-noise [30] | 12000 | 2000 | 10 |
| CIFAR-10 [29] | 50000 | 10000 | 10 |
| NORB [32] | 291600 | 58320 | 6 |
| Rectangles [30] | 1200 | 50000 | 2 |

**Single hidden layer architecture**  In these experiments, we used an architecture consisting of a fully-connected hidden layer, followed by a fully-connected softmax layer. In order to be consistent with the architecture used in Sindhwani et al. [45], we do not use a bias term in the hidden layer.

**CNN architecture**  In these experiments, shown in Table 7 in Appendix E, we tested on a LeNet-based architecture. The architecture has 2 convolution/pool layers with 6 and 16 channels respectively, followed by a fully-connected layer, followed by fully-connected logit/softmax layer. We replaced the second to last fully-connected layer, which was of dimensions $784 \times 784$ for the MNIST-bg-rot and MNIST-noise datasets, and $1024 \times 1024$ for the CIFAR-10 and NORB experiments.

**Replacing convolutional layers**  This experiment corresponds to Table 3.

Here, we investigated whether the convolutional layers of CNNs can be learned automatically. For our experiments, we test on the simplest possible multi-channel CNN model on the CIFAR-10 dataset. The model consists of one layer of convolutional channels (3 RGB in channels, 3 out channels, stride 5), followed by a fully-connected layer and a final FC+softmax layer (total of 4 layers). We replace the convolutions with various structured matrices of the same dimensions, keeping the same $3 \times 3$ channel structure (e.g. it would consist of $3 \cdot 3 = 9$ square structured matrices) and number of hidden units.[9]

The LDR classes benefit from being composed with LDR matrices of the same type (due to the composition property, Proposition 1(c)), so we additionally replace the later FC layer with the same structured matrix type.

By Proposition 1(d), channels of Toeplitz-like matrices form a larger Toeplitz-like matrix of the same size. Using this insight, we consider replacing the channel structure of the convolutional layer with either channels of structured matrices or a single wide structured matrix. (Also, note that this is able to leverage the asymptotic fast nature of our structured classes.)

Because it seems that convolutional layers are strongly dependent on pooling – our structured matrices outperform them in isolation – we compare against a version of the CNN with an additional pooling layer after the convolutional channels. Note that this comparison is the same basic four layer model with a structured matrix vs. a five layer convolutional model with pooling. Since the architectures are quite different and difficult to directly compare, we also experimented with adding more hidden units to the pooling model.

### F.2  Language modeling

For a language modeling application[10], we explored replacing weight matrices in a recurrent neural network with structured matrices. We evaluate on a single layer LSTM architecture, defined by the update equations:

$$
\begin{aligned}
i &= \sigma(W_{ii}x + b_{ii} + W_{hi}h + b_{hi}) \\
f &= \sigma(W_{if}x + b_{if} + W_{hf}h + b_{hf}) \\
g &= \tanh(W_{ig}x + b_{ig} + W_{hg}h + b_{hg}) \\
o &= \sigma(W_{io}x + b_{io} + W_{ho}h + b_{ho}) \\
c' &= f * c + i * g \\
h' &= o \tanh(c')
\end{aligned}
$$

In our experiments we replace the matrices $W_{ii}, W_{if}, W_{ig}, W_{io}$ with structured matrices. We use a hidden layer of size 128, and word embedding size of 128. We evaluate on the Wikitext-2 dataset, which consists of Wikipedia articles (2,088,628 training, 217,646 validation, and 245,569 test tokens). The total vocabulary is of size 33,278. We use the default hyperparameters and train using stochastic gradient descent with an initial learning rate of 20. The learning rate is annealed 4x after each epoch if performance does not improve on the validation set. Results are shown in Table 2.