[Reviews · NeurIPS 2018]

Reviewer 1



This paper proposed a method to reduce the number of parameters in deep neural networks with low displacement rank (LDR) matrices. It proposed a group of LDR matrices and several general displacement operators. Also, it proposed a method to learn the low-rank component/operator at same time and gave a bound on VC dimensions of multi-layer neural networks for structured matrices. The paper is well written although the terminology might be hard to follow for some readers. The method is a generalization of several previous works on LDR framework for DNNs and the proposed algorithm of learning low-rank component together is somewhat intuitive. However the theoretical analysis on complexity is original to my knowledge and the empirical study is extensive. Overall it is above average and I suggest for accept. Comments: Authors proposed many kinds of LDR matrices and gave a bound on VC dimensions with LDR matrices on neural networks, however, what is the advantage of learning both operator and the low-rank component at same time? Can you show the advantage with VC dimension? To my understanding, most training algorithms with LDR weight matrices are not scalable as they are at most up to BLAS level 2 operation. What's the computational efficiency with GPU of these different displacement operators? Detailed comments: -- Figure(2) sub-diagonal should be [x_1, x_2, ...]

Reviewer 2



# Post rebuttal Given new results and comparisons provided by the authors I am increasing initial score of (6) to (7). # Original review This paper focuses on an low displacement rank matrices used as parametrisations of neural networks and more precisely on defining a rich class of them, which can be used while preserving reasonable VC bounds. I find presented results interesting and valuable, however it is unclear how significant for the community they really are. Authors seem to view the approach as a way to decrease overparametrisation of the networks with minimal effect on accuracy. However, there are dozens of techniques which try to address this very issue, which are not really compared against. Instead, authors choose to focus on comparing to other, very similar approaches of reparametrising neural network layers with LDRs. Pros: - Clear message - Visible extension of previous work/results - Showing both empirically and theoretically how proposed method improves upon baselines - Providing implementation of the method, thus increasing reproducability Cons: - all experiments are relatively low-scale, and the only benefits over not constraining the structure is obtained for MNIST-noise (90 vs 93.5%) and CIFAR-10 (65 vs 66%) which are not very significant differences at this level of accuracy for these problems. Since this paper is not evaluation-centric, I am still in favour of accepting, but in a marginal sense; if provided with stronger empirical evidence for the method I would be keen to change the score. On the other hand there are many other ways of reducing number of parameters of a neural network, so if the evaluation claim is "minimisation of number of weights given small accuracy drop" than one should compare against exactly these approaches, such as: network prunning, even methods like "optimal brain surgeon" from 90s; micro-architectures which were found explicitely to minimise this constraint (like MobileNets etc.). - significance of these results are quite limited as the empirical evaluation shows limited benefits (or at least the proper evaluation metric, a number authors want to optimise, is unclear to the reviewer) and the theoretical results, while valuable, don't seem strong enough to consider it as a purely math paper. Minor remarks: - please fix language errors/typos, such as: Table 2 caption "has perplexity is", using "low rank" and "low-rank" (please pick one),

Reviewer 3



The paper introduces a rich class of LDR matrices and more general displacement operators enabling learning over both operator and low rank component with joint learning of operators and rank competent. The paper proves bounds on the VC dimension and sample complexity accompanied by empirical results on compression FC, convolutions and language modeling with significant performance improvements or matching performance on parameter budgets/fixed budget respectively. The authors also show the structured classes have lower generalization error. The paper clearly shows consistent higher test accuracy/matched accuracy than baselines replacing layers with the compressed operators. They show that learned operators have better accuracy over fixed operator They show by increasing rank of the unconstrained layers of the neural networks . Low sample complexity without compromising accuracy and sometimes outperforming with far fewer parameters. I think the originality lies in the bounds that the authors prove on VC dimension and sample complexity and the extensive evaluation on the significance and effect of the approximately equivariant linear map A,B that model high level structure and invariance while the low complexity remainder R control standard model capacity. The theory seems to work although I have not verified all the proofs in detail. It would be interesting to see empirical comparisons between the structured class of jointly learnable operator and rank LDRs versus other other compression techniques Hashnets, FastFood etc. In general, the paper is very well written and structured. The authors clearly state their important contributions and justify the claims with adequate theoretical guarantees and empirical results. The work is explained clearly and without ambiguity. A bit more background on related work on LDRs in the context of compression would be good.